# Using terrestrial laser scanning to constrain forest ecosystem structure and functions in the Ecosystem Demography model (ED2.2)

Félicien Meunier[1], Sruthi M. Krishna Moorthy[1], Marc Peaucelle[1*], Kim Calders[1], Louise Terryn[1], Wim Verbruggen[1], Chang Liu[1], Ninni Saarinen[2,3], Niall Origo[4], Joanne Nightingale[4], Mathias Disney[5,6], Yadvinder Malhi[7], and Hans Verbeeck[1]

[1]CAVElab - Computational and Applied Vegetation Ecology, Department of Environment, Ghent University, Ghent, Belgium

[2]Department of Forest Sciences, University of Helsinki, Finland

[3]School of Forest Sciences, University of Eastern Finland, Finland

[4]NPL - Climate and Earth Observation (CEO) group, National Physical Laboratory

[5]UCL Department of Geography, Gower Street, London WC1E 6BT, UK

[6]NERC National Centre for Earth Observation (NCEO), UCL Geography, Gower Street, London, WC1E 6BT, UK

[7]Environmental Change Institute, School of Geography and the Environment, University of Oxford, Oxford, UK

*now at INRAE, Université de Bordeaux, UMR 1391 ISPA, 33140 Villenave-d'Ornon, France

*Correspondence to*: Félicien Meunier (Felicien.Meunier@UGent.be)

**Abstract.** Terrestrial Biosphere Models (TBMs) are invaluable tools for studying plant-atmosphere interactions at multiple spatial and temporal scales, as well as how global change impacts ecosystems. Yet, TBM projections suffer from large uncertainties that limit their usefulness. Forest structure drives a significant part of TBM uncertainty as it regulates key processes such as the transfer of carbon, energy, and water between the land and the atmosphere, but remains challenging to observe and reliably represent. The poor representation of forest structure in TBMs might actually result in simulations that reproduce observed land fluxes, but that fail to capture carbon pools, forest composition, and demography. Recent advances in Terrestrial Laser Scanning (TLS) offer new opportunities to capture the three-dimensional structure of the ecosystem and to transfer this information to TBMs in order to increase their accuracy. In this study, we quantified the impacts of prescribing initial conditions (tree size distribution), constraining key model parameters with observations, as well as imposing structural observations of individual trees (namely tree height, leaf area, woody biomass, and crown area) derived from TLS into the state-of-the-art Ecosystem Demography model (ED2.2) at a temperate forest site (Wytham Woods, UK). We assessed the relative contribution of initial conditions, model structure, and parameters to the overall output uncertainty by running ensemble simulations with multiple model configurations. We show that forest demography and ecosystem functions as modelled by ED2.2 are sensitive to the imposed initial state, the model parameters, and the choice of key model processes. In particular, we show that:

- parameter uncertainty drove the overall model uncertainty with a mean contribution of 63% to the overall variance
of simulated gross primary production;
- model uncertainty on the gross primary production was reduced fourfold when both TLS and trait data were integrated
into the model configuration;
- land fluxes and ecosystem composition could be simultaneously and accurately simulated with physically realistic
parameters when appropriate constraints were applied to critical parameters and processes.
We conclude that integrating TLS data can inform TBMs on the most adequate model structure, constrain critical parameters,
and prescribe representative initial conditions. Our study also confirms the need for simultaneous observations of plant traits,
structure and state variables if we seek to improve the robustness of TBMs and reduce their overall uncertainties.

## 1 Introduction

Terrestrial biosphere models (TBMs) are key tools to understand the ecosystem response to anthropogenic disturbances and climate change (Medvigy and Moorcroft 2012; McGuire et al. 2001). Nowadays they are intensively used, as is or embedded in Earth system models, to study plant-atmosphere interactions and predict the future of ecosystems facing global change (e.g., Poulter et al. 2010). Yet, the usefulness of TBMs is currently limited by the large uncertainties in their projections which originate from different sources (Lin et al. 2011).

Forest structure has long been recognized as a critical component to understand forest dynamics (Hurtt et al. 2010). It influences the climatically important fluxes of carbon, energy, and water (Bonan 2008). Yet, its realistic representation is challenging and an urgent priority in the development of next-generation TBMs (Fisher et al. 2018). The representation of the forest structure within TBMs is associated with three sources of uncertainty: model structure, model initialisation, and model parameter uncertainty.

The model structure entails by definition all the processes included in a model, how they are implemented, and all the underlying assumptions (Bonan 2019). Model structure complexity varies among TBMs and also depends on the user configuration choices: different formulations of the same process can co-exist within a TBM. This complexity results from the necessary compromise between an accurate representation of reality on the one hand and the computational demand and observational requirements on the other (Shiklomanov et al. 2020). Model intercomparison studies have demonstrated that discrepancies in the representation of key processes such as forest structure (Fisher et al. 2018) or photosynthesis (Rogers et al. 2017) lead to significant uncertainties in the projections of critical variables such as the overall land carbon sequestration capacity (Friedlingstein et al. 2014; Lovenduski and Bonan 2017; Friedlingstein et al. 2006).

The initialisation uncertainty reflects the error made when determining the initial conditions of the modelled ecosystem. Several approaches exist for initialising TBMs, the most common of which is probably to start runs from near-bare ground conditions, force the simulations with relevant climate-forcings, and wait for the model to reach an equilibrium state, the so-called potential vegetation (Antonarakis et al. 2011). Yet, such a spin-up approach does not guarantee reliable initial demography, carbon pools, or ecosystem structure. Alternatively, forest inventories can be used to prescribe the initial composition of the ecosystem (Medvigy et al. 2009). The derivation of the initial states of critical variables, such as the aboveground biomass or the total leaf area from the plant size distribution, then relies on model default allometries which are often derived from other, potentially non-representative site-specific data.

Parameter uncertainty arises among other things from the necessary simplification of the natural complexity into a coherent list of model parameters, the uncertainty in the measurements used to calibrate the model, or the methods used to upscale local measurements to scales on which TBMs operate (Zaehle et al. 2005). Previous sensitivity analyses have underlined the critical importance of parameter uncertainty for the projections of ecosystem demography and productivity (Dietze et al. 2014;

Massoud et al. 2019; Raczka et al. 2018; Wramneby et al. 2008). In a recent comparative study, parameter uncertainty was even shown to dominate the overall model uncertainty over process uncertainty (Shiklomanov et al. 2020). Among model parameters, allometric coefficients scale the shape and mass of the plants or of its components with their size (Chave et al. 2014). Not surprisingly, multiple TBMs were shown to be sensitive to such allometric parameters (Collalti et al. 2019; Cano et al. 2020; Esprey et al. 2004). Parameter uncertainty can be reduced by constraining the range of variation of model parameters through the assimilation of different sources of observations or via model optimization (LeBauer et al. 2013). In the past, TBMs have often been calibrated with eddy covariance data (Fer et al. 2018; Rezende et al. 2016; Collalti et al. 2016). While this approach ensures that the model correctly reproduces the short timescale (diurnal/seasonal) dynamics of land fluxes, it does not ensure an accurate representation of forest structure and carbon pools. This is especially true because forest structure-related parameters can present a low sensitivity to those observations (LeBauer et al. 2013; Richardson et al. 2010), and the equifinality in TBMs (Luo et al. 2009) can lead to acceptable land fluxes with a poor representation of ecosystem structure (i.e. fluxes can be reproduced from an almost infinite range of structural possibilities, some of which will be much more likely than others).

Among the different sources of observations used to reduce model uncertainties, remote sensing from various platforms (terrestrial, air- and space borne) has increasingly been used to monitor and understand terrestrial ecosystems (Jones and Vaughan 2010). LiDAR (Light Detection And Ranging) data in particular have been used in the past to initialise forest biomass and constrain predictions of TBMS (Thomas et al. 2008; Hurtt et al. 2019). The recent revolution in Terrestrial Laser Scanning (TLS, also called terrestrial LiDAR) provides new opportunities for constraining TBMs, and reducing the uncertainties related to the vegetation structure representation (Fischer et al. 2019). The ability of TLS to measure the distance to reflecting surfaces was initially used in ecological studies to measure simple metrics like DBH and tree heights (Maas et al. 2008; Hopkinson et al. 2011). Since then, TLS methods have rapidly evolved to derive more complex metrics, such as the vertical profiles of the forest structure (Jupp et al. 2009; Calders et al. 2018) and whole-tree volumetric assessments (Fan et al. 2020), leading to an accurate determination of forest structure across various forest types (Calders et al. 2015; Tanago et al. 2018; Takoudjou et al. 2018; Ehbrecht et al. 2017; Stiers et al. 2018; Saarinen et al. 2021.). Today, the ability of TLS to accurately represent the 3D structure of forests via quantitative structure modelling (QSM), see Raumonen et al. (2013) and Hackenberg et al. (2015) represents a unique opportunity to improve our understanding of forest ecosystems under changing climates (Calders et al. 2020). In particular, TLS snapshots of vegetation ecosystems could simultaneously provide important state variables to initialise TBMs, strong constraints to some critical allometric parameters, and help determine the most appropriate model structure for some key processes.

In this study, we evaluated the relative contribution of different sources of uncertainty (parameters, processes, initial conditions) to the overall uncertainty of multiple simulated outputs of a specific TBM, namely the Ecosystem Demography model version 2 (ED2.2). We also explored the benefits of constraining vegetation structure related parameters and processes using TLS on the model performance and output variability. To do so, we ran ED2.2 simulation ensembles for a temperate

forest in the UK considering different initial states for the modelled ecosystem, and varying multiple model parameters and

process settings with or without TLS constraints. In other words, we assessed: (i) the relative importance of the model structure,

initialisation, and parameter uncertainties in the ED2.2 model representation of a temperate forest; (ii) the potential added

value of TLS data for vegetation modelling. To the best of our knowledge, this study is the first attempt to constrain a TBM

using TLS.

## 2 Material and Methods

### 2.1 Study site and data

#### 2.1.1 Study site

Wytham Woods is a mixed deciduous forest, predominantly broadleaved, covering approximately 40 ha. It is located 5 km northwest of Oxford in southern England (Thomas et al. 2011). Owned by Oxford University, Wytham Woods has been part of the UK Environmental Change Network (ECN) and of the Smithsonian Global Earth Observatory (SIGEO) network since 1992 and 2008, respectively, and has hosted numerous ecological studies (Savill et al. 2010). The site is classified as an ancient semi-natural woodland (Hall et al. 2001), which means that the site has been continuously covered by trees through recorded history (since at least 1600), occasionally managed, and experienced minimal intervention (i.e. no silvicultural management) since WWII (Fenn et al. 2015). Over the 1993-2008 time period, the site was characterised by a mean annual temperature of 10°C and a mean annual precipitation of 726 mm (Butt et al. 2009). The area we simulate in this study is a 1.4 ha forest plot nested within the 18 ha long-term monitoring site part of the ForestGEO global network of forest inventory plots. This 140 m × 100 m area has a local SW-coordinate (0, 100) and local NE-coordinate (140, 200) boundary. The local origin coordinate (0,0) was located with a differential GPS at Lat 51.7750579 and Lon -1.33904729.

#### 2.1.2 Field inventory and Terrestrial Laser Scanning data

The studied plot was inventoried during the summer of 2016. All trees were located, measured, and identified at the species level. The plot is largely dominated by sycamore (Acer pseudoplatanus, 65.3% of the 815 inventoried trees in the 1.4 ha plot, see Table 1, Figure 1 and Supplementary Figure S1), ash (Fraxinus excelsior, 10.3% of the stems), and hazel (Corylus avellana, 8.2% of the stems). Oaks (Quercus robur) represent a limited fraction of the woody stems (4.3%) but disproportionately contribute (23.4%) to the total basal area as they mostly consist of large trees (Table 1 and Figure 1). From the inventory, tree DBH is 24.4 cm on average (DBH median is 19.8 cm), and ranges from 2.9 cm to 141.2 cm.

Three-dimensional forest structure data were collected using a RIEGL VZ-400 terrestrial laser scanner (RIEGL Laser Measurement Systems GmbH) in leaf-on (June and July 2015) and leaf-off (December 2015 and January 2016) conditions (Calders et al. 2018). The RIEGL instrument uses on-board waveform processing and records multiple return LiDAR data, which improves vertical sampling (Lovell et al. 2003; Calders et al. 2014). Individual trees were extracted using treeseg (Burt, et al. 2019), and their structure modelled with TreeQSM (Raumonen et al. 2013) with the leaf-off TLS point cloud. Leaves were then added to the individual tree branches using both the leaf-off and -on TLS datasets with the FaNNI algorithm (Åkerblom et al. 2018). Doing so, TLS allowed retrieving of individual tree height, aboveground woody biomass (modelled through estimates of volume combined with species-specific wood density), and leaf area. In addition, the individual tree crown area was computed from the vertical projection of the leaf-off point clouds of individual trees. For more details, a complete description of the TLS data collection and forest stand reconstruction is available in Calders et al. (2018).

**140  2.1.3 Flux tower data and species traits**

Stand-scale carbon and water fluxes have been occasionally measured in Wytham Woods using the eddy covariance technique.
We digitised the most recent (to our knowledge) data collection of $CO_2$ fluxes that was reported by Thomas et al. (2011) for
the period May 2007-April 2009. To do so, we digitised the weekly mean values of ecosystem gross primary productivity
(GPP), ecosystem respiration ($R_{eco}$), and net ecosystem productivity (NEP) from Figure 6 of the aforementioned reference
using the Plot digitizer software (v.2.6.8, http://plotdigitizer.sourceforge.net/). For a more detailed description of the eddy
covariance data (including the data frequency of the original data, and the data quality filtering), we refer the readers to the
original publication by Thomas et al. (2011).
In addition, we extracted all existing records of specific leaf area (SLA) and maximum rate of carboxylation ($V_{c,max}$) for the
five most important species in Wytham woods (Acer pseudoplatanus, Corylus avellana, Crataegus monogyna, Fraxinus
excelsior, and Quercus robur) from the TRY database (Kattge et al. 2020), see Table 1 (the complete list of references from
which the data originate is available in supplementary section 1). Individual traits were converted into ED2.2 units (m² $kg_C^{-1}$
for SLA with a fixed leaf carbon content of 0.5 and $\mu mol_C$ m$^{-2}$ s$^{-1}$ for $V_{c,max}$). $V_{c,max}$ data were also rescaled to the ED2.2
reference temperature (15°C) using the model default value for the temperature coefficient Q10 of 2.4. Following Asner et al.
(2017), we calculated the community-weighted mean (CWM) and community-weighted standard deviation (CWSD) for both
traits based on the species composition and species-level average values, using species basal area as weights:

$$CWM = \frac{\sum_{i=1}^{N} w_i \cdot x_i}{\sum_{i=1}^{N} w_i}$$  Equation (1)


$$CWSD = \sqrt{\frac{\frac{\sum_{i=1}^{N} w_i \cdot (x_i - CWM)^2}{(N-1)\sum_{i=1}^{N} w_i}}{N}}$$  Equation (2)

where $N$ is the total number of species for which data was available in TRY for each trait $x$, $x_i$ is the mean trait value for
species $i$, and $w_i$ is the species weight (here the basal area of species $i$).
Flux tower data were used as a validation dataset while the TRY data were used to constrain parameters of the TBM used in
this study and described just below.
**2.2 Model**
**2.2.1 The terrestrial biosphere model ED2.2**
ED2.2 is a terrestrial biosphere model that can simulate the vegetation dynamics of a wide range of ecosystems from boreal to
tropical forests (Longo et al. 2019). It is a cohort-based, spatially implicit model that approximates the behaviour of an
individual-based, spatially distributed vegetation model through a system of size- and age-structured partial differential

equations (Moorcroft et al. 2001). ED2.2 integrates modules of plant growth, mortality, phenology, disturbance, hydrology, and soil biogeochemistry to predict e.g., the demography, the succession, and the dynamics of water and carbon within the simulated ecosystem.

In ED2.2, the inter- and intra-specific diversity is represented by a set of plant functional types (PFTs) that differ by their leaf physiology, phenology, growth and allocation strategies, mortality, and sensitivity to environmental conditions (D. Medvigy et al. 2009). The trees inventoried in Wytham Woods were classified as either mid- or late-successional temperate deciduous trees (see below for the reasoning of the mapping). These PFTs are cold-deciduous, i.e. leaf phenology is prognosed by the accumulation of growing degree-days (growing season) and chilling days (senescing season) (Longo et al. 2019). A comprehensive model description, including photosynthesis, allometries, radiative transfer, and phenology, is available in Longo et al. (2019).

**2.2.2 Model initialisation and forcings**

In this study, the ED2.2 model was initialised using i) near-bare ground (NBG) initial conditions (i.e. seedlings only), ii) the field inventory, or iii) the TLS-reconstructed size distribution. In the latter two configurations, the 1.4 ha site was initially divided into 35 square patches of 20 x 20 m. These three types of initial conditions are referred to below as NBG, Census, and TLS respectively. Simulations were run for multiple years using the local forcing data of the corresponding years of the CRU-NCEP reanalysis dataset (Viovy 2018). Simulations were run for either five years (Census and TLS configurations) or the approximate age since the last large-scale disturbance (100 years, NBG configuration), see Table 5. Soil texture was set according to the dominant soil type (clay), based on site-level observation (Butt et al. 2009).

**2.2.3 Allometries and model parameters**

In ED2.2, the carbon made available from net assimilation is partitioned at the cohort level into the different plant pools according to DBH-dependent allometries (Longo et al. 2019). In other words, plant cohorts allocate the carbon assimilated through photosynthesis to living tissues (i.e. fine roots, sapwood, leaves, seeds), the non-structural storage pool, and the dead tissues (i.e. coarse roots, and aboveground woody biomass) depending on (i) a set of allometries and (ii) whether the plant carbon balance and environmental conditions are favourable for growth. In ED2.2, aboveground woody biomass, height, leaf biomass, and crown area are scaled through DBH-dependent allometries (Table 3). The ED2.2 default allometric models and parameters are defined according to Medvigy et al. (2009) for the leaf biomass and height, Dietze et al. (2008) for the crown area, and Albani et al. (2006) for the aboveground woody biomass.

To estimate the relative contribution of the parameter uncertainty to the variability of the model outputs, we used parameter distributions from previous ED2.2 parameter uncertainty studies (Dietze et al. 2014; Shiklomanov et al. 2020; Raczka et al. 2018; Viskari et al. 2019). We only targeted those parameters that were shown to significantly contribute to the overall

parameter uncertainties in the aforementioned studies (Table 4) and set the rest to their ED2.2 default values for all simulations.
For SLA and $V_{c,max}$ in particular, we defined two types of parameter distributions: either relatively wide priors as in the previous
sensitivity analyses listed above (Table 4) or constrained posteriors generated by the trait meta-analysis of the Predictive
Ecosystem Analyzer (PEcAn) run with the existing data in TRY and without random effects, see (LeBauer et al. 2013; Meunier
et al. 2021; Raczka et al. 2018). The meta-analysis was informed by TRY data only. Those distributions are referred to below
as without or with TRY-constraints, respectively. The uncertainty of the allometric coefficients was determined either by the
range of variation of those parameters in the ED2.2 model for hardwood tree PFTs (NBG and Census configurations) or by
the posterior distributions of these parameters generated when fitting the TLS data (see below).

## 2.2.4 Model configurations

To assess the importance of the model structure uncertainty, we targeted processes that were shown to induce significant
variability in the model outputs in previous studies (Shiklomanov et al. 2020). In detail, we ran the model with multiple
combinations of the following configurations: (i) closed canopies versus crowns of finite radii; (ii) two-stream versus multiple-
scatter canopy radiative transfer models (RTMs); (iii) static versus plastic (varying with available light level) SLA and $V_{c,max}$;
and (iv) a single versus two plant functional types (Table 2).
By default in ED2.2, plant canopies are represented as infinitely thin flat crowns (a.k.a. complete shading or closed canopy)
that virtually occupy the entire horizontal space of the patch in which the cohort is located. In an alternative configuration,
cohorts are still stacked on top of each other but have a finite radius and hence tallest plants only partially shade the underlying
cohorts. In other words, the crown sub-model of ED2.2 determines the nature of the light competition between cohorts. Closed
canopies have been shown to dramatically suppress competition from sub-dominant PFTs and typically result in unrealistically
homogeneous patches (Fisher et al. 2015) while understorey cohorts receive more incoming diffuse and direct light if finite
crowns are simulated.
The second sub-model we investigated was the choice of RTM. In both options (two-stream and multi-scatter), the full vertical
radiation profile within each patch is resolved as a function of the canopy structure (e.g. leaf and wood area, clumping) and
the environmental conditions (e.g. incident solar radiation, solar angle) following the approach of CLM 4.5 (Oleson et al.
2013). Both RTMs differ in the numerical resolution of the radiative transfers. By default (two-stream), the special multi-
canopy solution of the two-stream approximation for vegetation canopies (Sellers 1985) is used as described in Longo et al.
(2019) while the multiple-scatter is derived from first principles by Zhao and Qualls (2005) to address the long-known issues
and biases of the two-stream model (Wang 2003). The multiple-scatter configuration increases diffuse light levels in the
understorey as compared to the default two-stream approach (Shiklomanov et al. 2020).
The third sub-model that we evaluated is related to trait plasticity. By default (static), all cohorts of a given PFT share the same
set of parameters which do not evolve over time, in contradiction with well-documented intra-specific variability of plant traits
with environmental conditions (e.g. Keenan and Niinemets 2016). In the alternative configuration (plastic), cohort SLA and
$V_{c,max}$ respectively decrease and increase with light availability, following empirical relationships from the tropics (Lloyd et
al. 2010).
Finally, we also evaluated the impact of simulating one or multiple PFTs by either classifying all trees in the Wytham Woods
inventory as belonging to the mid-successional hardwood tree PFT of ED2.2 ($N_{PFT} = 1$) or according to a classification similar
to the one of Dietze and Moorcroft (2011), ($N_{PFT} = 2$), supplemented by a clustering analysis of the allometric relationships
derived from the TLS data (see below).
**2.3 Analyses**
**2.3.1 Impact of TLS data on model allometries and initial conditions**
We first compared the model default allometries with site-specific ones constrained from the TLS data. To do so, we fitted the
individual plant metrics (height, crown area, aboveground woody biomass, and leaf area) versus DBH relationships derived
from TLS with the set of equations used in ED2.2 (Table 2). More specifically, we fitted the parameters of the four allometries
of ED2.2 using a Bayesian approach and the 'brms' package of R (Bürkner 2017). To account for the uncertainty of the data
we repeated the same analysis multiple times ($N = 100$) using data random sampling with replacement and aggregating the
resulting allometric parameter posterior distributions. To convert the leaf area obtained from TLS into leaf biomass, we used
the CWM of SLA. We evaluated the quality of fit of the allometric models by computing the root-mean-square deviations
(RMSD, van Breugel et al. 2011) normalised by the observed mean and the Watanabe information criterion (WAIC) for all
four allometric models (height, crown area, aboveground woody biomass, leaf biomass). We fitted all allometric models using
multiple possible species-to-PFT classifications and only retained the classifications that minimised the WAIC for the
configurations $N_{PFT} = 1$ and $N_{PFT} > 1$.
To assess the relative importance of TLS for the model initialisation, we compared the tree size distributions obtained from
the field inventory and the TLS data and computed the absolute and relative differences between both DBH distributions
(ground-truthing of TLS).
**2.3.2 Ensemble runs**
For each type of initial conditions (NBG, Census, and TLS), we ran ensembles of 500 simulations with parameters randomly
sampled from the parameter distributions (Table 4) and with process configuration randomly selected from the different options
(Table 5). Each ensemble was equally split between runs with (250) and without (250) TRY constraints on SLA and $V_{c,max}$.
The same parameter samples and process configurations were used for all three types of initial conditions, and with and without
TRY restrictions on SLA and $V_{c,max}$ to allow independently evaluating the impact of the initial conditions, TRY- and TLS-
constraints at specific parameter values.

### 2.3.3 Sensitivity analyses and variance decomposition

Finally, we assessed which processes and parameters contributed the most to the overall model variance by performing a sensitivity and a variance decomposition analysis following Dietze et al. (2014) and Lebauer et al. (2013). This analysis allows predicting the fraction of the variance in target output variables attributable to individual parameters and processes (or "partial variance"). We chose as target output variables the ecosystem GPP during the most productive month (June) or over the leaf-on season (May-October), the total leaf area index (LAI) and the understorey photosynthetically active radiation (PAR) in leaf-on conditions, as well as the aboveground woody biomass at the end of the simulation. For the NBG configuration, we also decomposed the variance of the total stem density (which is prescribed in the other two configurations). Parameters included in the variance decomposition analyses were re-classified as belonging to one of these three categories: allometric parameters, TRY-constrainable parameters (SLA and $V_{c,max}$), and others. All five years of the Census and TLS configurations were kept for analysis while only the last five years of the NBG runs were considered. Note that the variance partitioning algorithm that we used only attributes to the parameters and processes their direct effect: interactions are not accounted for in the variance decomposition.

All analyses presented in this study were performed using R 3.6 (R Core Team, 2019).

## 3 Results

### 3.1 Impact of TLS data on model allometries and initial conditions

TLS-extracted and field inventory DBHs were very well correlated ($R^2$ = 0.98, slope of the inventory vs TLS linear model = 0.998, see supplementary Figure S4). The mean (resp. median) relative difference between the TLS and field inventory DBHs was -0.2% (resp. -1.7%), see Supplementary Figure S5. The minimum and maximum absolute differences in DBH were -13.8 and 32.9 cm, respectively; the minimum and maximum relative differences were -42 and 101%, respectively (Supplementary Figure S5). The total tree basal area from the inventory was 36.8 cm$^2$ m$^{-2}$ while the total tree basal area obtained from TLS tree reconstruction was 36.2 cm$^2$ m$^{-2}$.

Individual tree measurements from QSMs applied to the TLS point cloud could all be satisfactorily represented by the ED2.2 allometric equations and a single PFT (Figure 2). $R^2$ of the allometric models for the individual aboveground woody biomass, height, crown area, and leaf biomass respectively reached 0.95, 0.83, 0.67, and 0.77. The normalised RMSD changed from 18.3 to 16.9% (height), from 85.1 to 75.7% (crown area), from 146.1 to 95.0 (woody biomass), and from 151% to 83.5% (leaf biomass) when switched from ED2.2 default allometries for the mid-successional hardwood tree PFT to TLS-derived, site-specific ones (Table 3).

Over the DBH range in Wytham Woods, TLS−derived allometries led to systematically larger allocations to aboveground woody biomass (+73% on average, up to +177% for the smallest tree) and leaf biomass (+75% on average), and smaller tree height (−1.9 m on average) as compared to ED2.2 defaults (Figure 2). Individual crown areas derived from TLS measurements varied between 0.2 and 465.4 m², with a mean of 26 m². As compared to the TLS−calibrated allometries, default model coefficients predicted larger crown areas for trees with DBH < 64 cm (−22% on average), and smaller crown areas for trees with DBH ≥ 64 cm (+17% on average), see Figure 2. The latter category (DBH ≥ 64 cm) comprised 30 trees (3.7% of the total) and contributed to 30.7% of the total basal area and 24.9% of the total leaf area.

Increasing the number of PFTs only slightly improved the goodness of fit of all four allometric models. The best species-to-PFT mapping according to the literature-informed minimization of the Watanabe information criterion was to classify Acer pseudoplatanus as belonging to the late-successional hardwood PFT and the rest of the tree species as belonging to the mid-successional hardwood PFT (Table 1, Supplementary Figures S2 and S3). Using this classification, the normalised RMSD of the allometric models decreased from 16.9 to 16.8% (height), 75.7 to 71.1% (crown area), 95.0 to 77.9% (aboveground woody biomass), and 83.5 to 73.9% (leaf biomass). This mapping resulted in larger crown areas and larger carbon allocation to woody

and leaf tissues for small (DBH < 50 cm) trees of the mid-successional tree PFT and taller late-successional trees across all
DBHs (+1.16 m on average).

**3.2 Ensemble runs**

Regardless of the TRY constraints and the initial conditions, the model ensembles could on average reproduce both the
amplitude and the seasonality of the gross ecosystem productivity, as observed by the eddy covariance flux tower, with a
maximum GPP in June and a leaf-off season with close-to-zero GPP in December-February (Figure 3). $R^2$ of observed vs
simulated monthly mean of GPP was larger than 0.93 for all configurations (NBG, Census, TLS) while the RMSE varied
between 1.2 (NBG), 1.3 (TLS) and 1.9 (Census) µmol m$^{-2}$ s$^{-1}$, much lower than the mean and standard deviation of the two
years of observational data of GPP (5.5 and 4.7 µmol m$^{-2}$ s$^{-1}$, respectively). Because we only simulated fully deciduous tree
PFTs, model ensembles underestimated GPP during winter: simulated ecosystem LAI and hence ecosystem gross productivity
dropped to almost zero in December-February (Supplementary Figure S6) while measured ecosystem productivity was non-
null during the same period (Figure 3), driven by evergreen understory plants such as shrubs that were not included in our
simulations.
The variability of the simulated GPP was critically influenced by the model configuration and the application of constraints
on SLA and $V_{c,max}$ (Figure 3). The standard deviation of the ensemble runs for the simulated GPP was not unexpectedly the
largest for the configuration with the least information on the ecosystem (the NBG configuration without TRY constraints),
and reached 6.33 µmol m$^{-2}$ s$^{-1}$ for June (Figure 3). More than 23% of the runs in that configuration led to unvegetated conditions
(LAI < 0.1 m$^2$ m$^{-2,}$ all year long, see Supplementary Figure S6) after 100 years of simulations while about 5% of the runs
simulated unrealistically dense tree covers (LAI > 10 m$^2$ m$^{-2}$ in summer). Combined with the uncertainty of all other
parameters, including photosynthetic ones, the LAI variability explains the extreme variability of the simulated ecosystem
gross productivity. The 95% confidence interval of the simulated ecosystem GPP in June for the NBG configuration without
TRY constraints (0 - 19.8 µmol m$^{-2}$ s$^{-1}$) was almost twice as large as the observed GPP at that moment (13.2 µmol m$^{-2}$ s$^{-1}$).
Prescribing initial conditions reduced the variability of the simulated outputs: ensemble standard deviation of GPP in June for
the Census configuration without TRY constraints was 4.83 µmol m$^{-2}$ s$^{-1}$. However, for the ecosystem productivity constraining
SLA and $V_{c,max}$ was even more critical: ensemble standard deviation of GPP in June for the Census configuration with TRY
constraints decreased to 1.99 µmol m$^{-2}$ s$^{-1}$ (see Figure 3 and also Figure 4 where the pie chart radius is set proportional to the
variance of the simulated ecosystem GPP). When both parameters were constrained and realistic initial conditions were
prescribed to the model (i.e. going from the NBG-without TRY constraints to the Census-with TRY constraints configuration),
the variability of the simulated GPP experienced a three-fold decrease. Similarly, the variability of LAI (supplementary Figure
S6-7) and AGB (supplementary Figure S8) was drastically reduced, with a four-fold and and a two-fold decrease respectively.

Given the similarities of the tree size distributions derived from the inventory and TLS (see results section 3.1), prescribing initial conditions had a similar impact on the variability of the outputs for the TLS and for the Census configurations. Combined with the constraints on allometries, it led to a reduction of the ensemble standard deviation for GPP in June to 3.78 µmol m$^{-2}$ s$^{-1}$ for the TLS configuration without TRY constraints. As for the Census configuration, constraining SLA and $V_{c,max}$ with TRY data had a larger impact on the model uncertainty: ensemble standard deviation of GPP in June for the TLS configuration with TRY constraints decreased to 1.54 µmol m$^{-2}$ s$^{-1}$. Incrementally adding the TLS-related information to the Census-with TRY constraints configuration had a positive, yet more limited effect on the reduction of the model variability of GPP: ensemble standard deviation of GPP in June was reduced by 30% between the Census and TLS configurations with TRY constraints. Constraining allometries with TLS had a more significant impact on LAI (supplementary Figures S6-S7) and AGB (supplementary Figure S8), with a three-fold decrease of the ensemble standard deviation from the Census-with TRY constraints to the TLS-with TRY constraints configurations.

All in all, the predicted variability of the ecosystem LAI and GPP was the lowest for the TLS configuration with TRY constraints: $3.79 \pm 0.50$ m$^2$ m$^{-2}$ for the ensemble mean ($\pm$ one standard deviation) of the ecosystem LAI (Supplementary Figure S6), $9.86 \pm 2.89$ µmol m$^{-2}$ s$^{-1}$ for the ensemble mean ($\pm$ one standard deviation) of the ecosystem GPP (Figure 3), both during leaf-on conditions, which compared well with independent observations (Table 6). The confidence interval of the simulated ecosystem GPP in June for the TLS configuration with TRY constraints was significantly reduced (11.8 - 17.6 µmol m$^{-2}$ s$^{-1}$) and much closer to the confidence interval of the observations (11.5 - 14.6 µmol m$^{-2}$ s$^{-1}$). In total, the variability of the simulated GPP experienced a four-fold decrease when parameters were constrained, realistic initial conditions were prescribed, and TLS data were used to constrain the allometries (i.e. going from the NBG-without TRY constraints to the TLS-with TRY constraints configuration).

**3.3 Variance decomposition and sensitivity analyses**

The variance of the ecosystem GPP was dominantly driven by the parameter uncertainty regardless of the configuration and the application of TRY constraints (Figure 4). Together, TRY-constrainable parameters, allometric coefficients, and the other ED2.2 parameters included in the sensitivity analysis, contributed on average to 63% of the total variance of GPP in June. Constraining SLA and $V_{c,max}$ with TRY datasets dramatically decreased the relative contribution of these two parameters to the overall variance: moving from uninformed priors to posteriors generated by the trait meta-analysis of PEcAn made the sum of their partial variances drop from a majority (57% on average for all three configurations) to a small contribution (7% on average for all three configurations), their share being mainly replaced by unconstrained parameters which increased from 6% to 50% on average across all configurations (Figure 4), especially the Quant. Eff., the Clumping and the Growth resp. parameters (Figure 5). The variance decomposition of the simulated ecosystem LAI and aboveground biomass led to very similar results, yet with a larger contribution of allometric parameters: allometric parameters contributed on average to 6 and

20% of the variance for LAI and AGB respectively, a larger contribution than theirs for the variance of GPP (3%), which
illustrates the importance of TLS to constrain the ecosystem structure (Figure 5 and Supplementary Figures S7-S8).
On average, processes only accounted for 12% of the overall variance of GPP with a maximum (resp. minimum) for the TLS
configuration with TRY constraints with 20% (resp. for the NBG without TRY constraints with 5%). Process uncertainty was
dominated by the type of crown model (5%) and the radiative transfer model (4%). Trait plasticity only contributed marginally
to the overall variance (< 1% on average). Processes (especially the choice of the RTM) played a stronger role for the available
light in the understorey (on average 40% of the total variance), especially in runs with prescribed initial conditions (on average
56% of the total variance, see Supplementary Figure S9). Due to compensatory effects (Supplementary Figure S2), the number
of simulated PFTs had a limited impact on all of the considered model outputs: $N_{PFT}$ only contributed to 3% of the variance of
ecosystem GPP, 2% of the variance of LAI and PAR, and 1% of the variance of AGB.

### 3.4 Ecosystem structure and functions

Despite similar seasonal cycles of ecosystem productivity (Figure 3), ensemble means exhibited very contrasted ecosystem
structure (Figures 6-7). None of the unprescribed simulations (NBG configuration) could capture the size distribution observed
through the inventory (Figure 6). Small-size stem (especially DBH < 50 cm) densities were underestimated while large tree
(DBH > 100 cm) densities were overestimated in the vegetated simulations (LAI > 0.1 $m^2 m^{-2}$) of the NBG configuration with
or without TRY constraints. Switching from closed canopy to finite crowns systematically increased the density of small (DBH
< 50 cm) trees, by 73% on average; just like constraining SLA and $V_{c,max}$ with TRY data. While the ecosystem LAI of the
NBG configuration with closed canopies compared well with independent observations from the literature (3.83 ± 1.94 versus
the range of 3.6 - 4.1 $m^2 m^{-2}$ observed in Wytham Woods, Table 6), the vertical arrangement of the leaves significantly differed
from what was observed by TLS and imposed in the TLS configuration (Figure 7), as a result of the differences in tree size
distribution (Figure 6).
Despite lower total leaf areas, the infinitely wide crown configuration (closed canopies, Table 6) made the forest more opaque
to the incoming solar radiation than the finite crowns. Across all configurations, the PAR available in the understory decreased
by 15% throughout the year while the ecosystem LAI decreased by 18% when closed canopies were simulated (Table 6). For
near bare-ground configurations, the LAI of the potential vegetation simulated was 23% lower with infinite crowns, and 16%
less PAR reached the understorey.
As the soil received more radiation when finite crowns were simulated, it was warmer and as a result, heterotrophic (and
ecosystem respiration, see Table 6) increased (+ 25% on average) when switching from infinite to finite crowns. Forest carbon
stocks also diverged between configurations: driven by higher allocations to leaf and aboveground woody biomass (Figure 2),
aboveground carbon storage was larger (+74% on average) in TLS-derived runs than when default allometries were applied
(Table 6). Aboveground woody biomass from configurations starting from near bare-ground conditions was systematically

underestimated compared to the TLS estimates (11.4 $kg_C$ m$^{-2}$ on average for the NBG configuration versus 24.5 $kg_C$ m$^{-2}$ on average for the TLS configuration). However, the larger allocation to woody biomass induced by the use of TLS-derived allometries mostly did not impact any other model outputs (Figure 5) as that carbon pool is inert and does not influence a lot of processes downstream (e.g. more woody biomass does not translate into exacerbated light interception). Leaf biomass allometry derived from TLS both reduced the simulated LAI and ecosystem GPP to more realistic values and constrained its variability (Figures 3, Table 6, and Supplementary Figure S6).

None of the simulation/configurations could accurately represent all features of Wytham woods. The model simulations starting from near bare-ground conditions failed to capture the vertical distribution of leaves (Figure 6) and the tree size distribution (Figure 7); the model simulations prescribed with the inventory overestimated the ecosystem GPP (Table 6); and the model simulations from the three configurations all overestimated the net ecosystem productivity (NEP), due to an overestimation of GPP (Census) and/or an underestimation of the ecosystem respiration (Census, NBG, and TLS), see Table 6. Model simulations underestimated $R_{eco}$ on average by -17% leading to unrealistic NEP predictions, which illustrates the need for constraining or optimising autotrophic and heterotrophic respiration parameters along with the photosynthetic and allometric parameters to align those with observational data.

## 4 Discussion

### 4.1 The relative weight of the different sources of uncertainty

The different model configurations tested in this study led to contrasting predictions of vegetation states. Depending on the chosen model outputs, the relative weights of the sources of uncertainty considerably varied. Near bare-ground simulations generated potential vegetations that significantly differed in their demography from observations (Figure 4) while prescribing initial tree size distribution was not a guarantee for accurately reproducing observed land fluxes (Figure 3, Table 6). The finite crown area representation also had a substantial impact on the model outputs. In particular, limiting the crown radius to finite values promoted smaller plants in the understorey (Figure 6), increased the simulated LAI (Table 6) and profoundly modified the vertical distribution of light in the canopy (Figure 8 and Table 6). Carbon pools also considerably diverged between model configurations, especially when TLS-derived allometries were taken into account (Table 6).

However, in general, it was the parameter uncertainty that dominated the overall model uncertainty (Figure 3, Supplementary Figure S7 and S8), just like it was previously observed for ED2.2 simulations of temperate forests (Shiklomanov et al. 2020). The parameters that dominated the variance depended on the use of TRY and/or TLS constraints. When observations were available, uncertainty was transferred to other unconstrained parameters while the overall variance was reduced, like in previous similar studies (Meunier et al. 2021), which supports the process of progressively integrating observations of most sensitive parameters until the model variance is reduced to satisfactory levels in an efficient data-model fusion loop (Dietze et al. 2014).

Although parameter uncertainty was larger in magnitude than process uncertainty, crown size representation and the choice of RTMs appear to drive a significant part of the model process uncertainty and should be paid more attention to in future analyses. Especially, because the implementation and the sensitivity of the radiative transfer processes are currently overlooked in ED2.2 like other vegetation models (Fisher et al. 2018; Viskari et al. 2019).

### 4.2 The added value of TLS for vegetation modelling

The quantitative information that remote sensing generates at unprecedented spatial and temporal scales can serve the purpose to reduce uncertainties in TBM projections. It has already been shown that airborne laser scanning (ALS) combined with an individual-based forest model could offer new insights into the contribution of plant size to ecosystem functioning (Fischer et al. 2019). Similarly, ALS and synthetic-aperture Radar have successfully been applied to prescribe the initial structure and composition of tropical forests (Antonarakis et al. 2011; Antonarakis et al. Moorcroft 2014; Longo et al. 2020), and LiDAR data have been coupled to allometric models to estimate carbon stocks and fluxes at large scale (Hurtt et al. 2019; Thomas et al. 2008). Yet, our study is the first attempt to inform a TBM with TLS data. As compared to ALS, TLS offers a few significant advantages, as well as some drawbacks, that are important to remember. Airborne techniques allow for wall-to-wall coverage characterising 3D forest structure at the regional scale, whereas TLS offers far more detailed information but only at the local

(up to a few ha) scale. However, TLS is capable of estimating the volume of individual trees directly, instead of relying on
allometries that require calibration and thus field measurements. In addition, it can accurately capture the entire size distribution
(DBH and height) of the sample plot while smaller trees can easily be missed with airborne surveys (Wang et al. 2016) leading
to incorrect demography, especially in dense forests.
Because TLS data are complementary to the datasets that are frequently used for model calibration (e.g. eddy covariance data),
they can contribute in a collective effort towards realistic representations of ecosystems in TBMs. TLS has the potential to fill
important parameter and process gaps and in doing so, to help reduce the uncertainties in vegetation model simulations. The
steep increase in the amount of available forest TLS data over the past decade (Calders et al. 2020) makes its coupling with
TBMs even more timely. As demonstrated in this study, TLS observation can ensure a more adequate model structure,
constrain model allometric parameters and prescribe representative initial conditions. Yet, only a combination of constraints
on both allometries (using TLS data) and photosynthetic parameters (thanks to TRY data) could satisfactorily reduce the model
uncertainties to its lowest level, which supports the integration of multiple data sources into TBMs for more realistic
simulations (Peylin et al. 2016). Such a combination of a TBM and multiple data streams allowed us to accurately simulate
both ecosystem productivity and ecosystem community composition with physically realistic parameters, which was
previously highlighted as a challenge for dynamic vegetation models (Shiklomanov et al. 2020; Fisher et al. 2010).
In the future, TLS could inform vegetation models even more. The TLS community is indeed actively working on the
derivation of additional tree- or stand-scale parameters from lidar raw data and 3D point clouds. Those parameters include leaf
angle distributions (Boni Vicari et al. 2019), clumping (Zhao et al. 2012), and reflectance (Calders et al. 2017), which have
been shown to significantly contribute to the overall model uncertainty (Meunier et al. 2021; Shiklomanov et al. 2020; Viskari
et al. 2019). Yet, theoretical, technological, and technical challenges specific to each parameter still need to be raised before
one can constrain these sensitive traits with TLS in a study similar to this one.
**4.3 Model equifinality**
Some runs from all three configurations (prescribed or not with initial size distributions) could reproduce the seasonal cycle
of GPP observed by the flux tower (Figure 3). However, those 'optimal' simulations were very different from the forest
structure point of view (Table 6, Figures 6-7). This situation illustrates the low identifiability of numerous TBM parameters
and the need for multiple simultaneous constraints and observations. While aboveground carbon storage is critical to estimate
forest sink strength and the overall carbon storage capacity of the ecosystem (Keeling and Phillips 2007), it has a limited
impact on simulated land fluxes (GPP in particular, see Figure 5) that are often used to calibrate TBMs. The parameters
controlling land fluxes, namely those controlling ecosystem LAI (Williams and Torn 2015; Wei et al. 2013) and those related
to photosynthesis (Figure 5), are also confounded, echoing observed trade-offs of the Leaf Economic Spectrum (Wright et al.
2004; Peaucelle et al. 2019). TLS has the potential to discriminate equifinal model simulations with similar land fluxes but
contrasting structure. On-site trait measurements (Figure 3) could further help avoid those risks of equifinality (Babst et al.
2020; Peaucelle et al. 2019).

**4.4 Study limitations**

Our findings come with several important limitations. First, the eddy covariance flux data (2007-2009) preceded the
observation of the forest structure (TLS and field inventory occurred over the 2015-2016 period) by almost a decade. The
forest composition and demography might have changed in the meanwhile, which reduces the confidence of the validation
with eddy covariance data (Figure 3). This is even more true as one realises that the validation dataset is rather limited in size
and information content (very low year-to-year variability in observed fluxes). Yet, in this study we were more interested in
the variance decomposition for different model configurations (Figures 3-4) than the actual goodness of fit of every single
configuration. In addition, in the absence of locally observed meteorological drivers, we had to force the model simulations
with regional datasets that cannot serve the purpose of capturing the day-to-day variability or the diel cycle, which forced us
to only compare the modelled and observed seasonal GPP cycle. Furthermore, GPP is not directly observed but rather a derived
(modelled) quantity as opposed to the net ecosystem exchange of carbon and the latent heat flux of water that are directly
measured. We could not access water flux raw data nor were they reported in publications that we knew of. GPP uncertainties
were also not quantified in the original publication of Thomas et al. (2011). While NEP values were reported, validating the
model simulations with those values would have biassed our analyses as we could not constrain respiration parameters with
data. Mismatches between different data sources and/or the low availability of good-quality data are recurrent issues in
vegetation modelling exercises. Despite multiple initiatives to standardise high quality data such as Fluxnet (Baldocchi et al.
2001), we emphasise here the need for concomitant observations in experimental and observational plots.
Second, the comparison between the potential vegetations as simulated by ED2.2 and the field inventory data are also imperfect
as Wytham Woods is a managed forest that has been frequently coppiced and pollarded. Disturbance history experienced by
the ecosystem is mostly unknown, preventing us from reproducing the current forest demography by the model.
Third, the trait meta-analysis was run with random effects turned off, which can generate too narrow parameter posterior
distributions (Raczka et al. 2018), and hence underestimate the contribution of the TRY-constrained parameters (see e.g. Figure
4). A similar analysis including random effects should be repeated to evaluate such an underestimation.
Finally, the ecosystem growth form complexity was neglected in this study. We only simulated tree PFTs while shrubs and
grass species also coexist in Wytham Woods. Integrating this ecological complexity would not have brought additional
information or robustness regarding the objectives of our study on the variance decomposition while increasing the
dimensionality and complexity of the problem. Future research should investigate whether the main findings highlighted in
this study hold with other PFTs, across other sites and biomes, or even in other vegetation models (Dokoohaki et al. 2021).

## 5 Conclusion

Vegetation models are important tools to predict the fate of ecosystems in a changing climate but are often used as black-box tools due to their complexity. They have been designed to realistically represent the ecosystem that they simulate, but often fail to do so primarily because of considerable parameter uncertainties as well as process and initialisation errors. Even for the state-of-the-art process-based terrestrial biosphere models, not all parameters can be constrained with data: some cannot be observed in the field, require calibration, or the appropriate observational trait data may be missing. In addition, model initialisation and the choice of model structure necessarily lead to additional uncertainties. We demonstrate in this study that TLS has the potential to provide initial condition estimates and to constrain some critical vegetation model parameters (allometries) and processes (crown representation). Combined with trait-based constraints on a few key parameters, TLS was able to define a model configuration that could reproduce both the ecosystem productivity and the plant community composition of the simulated site with physically realistic parameters, as well as considerably reduce model uncertainties.

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

**Code and data availability**

Code and supporting data (including initialization and setting files) for reproducing the results presented below are publicly available in Zenodo and have the permanent DOI 10.5281/zenodo.6363617. The ED2.2 model is available at https://doi.org/10.5281/zenodo.3365659.

**Acknowledgements**

This research was funded by BELSPO (Belgian Science Policy Office) in the frame of the STEREO III programme – project 3D-FOREST (SR/02/355). The computational resources and services used in this work were provided by the VSC (Flemish Supercomputer Center), funded by the Research Foundation - Flanders (FWO) and the Flemish Government – department EWI. During the preparation of this manuscript, F.M. was funded by the FWO as a junior postdoc and is thankful to this organisation for its financial support (FWO grant n° 1214720N). N.S. was funded by the Academy of Finland (project number 315079). K.C was funded by the European Union's Horizon 2020 research and innovation programme under the Marie Sklodowska-Curie grant agreement N° 835398. M.P. was funded by the FWO (grant No. G018319N) and the European Union's Horizon 2020 research and innovation programme under the Marie Sklodowska-Curie grant agreement No. 891369. The TLS fieldwork was funded through the Metrology for Earth Observation and Climate project (MetEOC-2), grant number ENV55 within the European Metrology Research Programme (EMRP). The EMRP is jointly funded by the EMRP participating countries within EURAMET and the European Union. Funds for purchase of the UCL RIEGL VZ-400 instrument was provided by the UK NERC National Centre for Earth Observation (NCEO). The census of the forest plot was supported by an ERC Advanced Investigator Grant to Y.M. (GEM-TRAIT, grant number 321131). We are grateful to the whole PEcAn group and the ED2 team for helpful discussions and support related to the functioning of PEcAn and ED2.

## Tables

Table 1: Mean (± one standard deviation) of plant traits (Specific Leaf Area or SLA, and maximum rate of carboxylation or $V_{c,max}$) available in the TRY database for each of the five dominant species in Wytham woods, and their local prevalence (in terms of individual density and basal area). Missing traits were unavailable in TRY. The table also summarises the abundance of those five dominant species in the 1.4 ha plot in terms of absolute and relative density and basal area, as well as the PFT mapping when more than one PFT were simulated ($N_{PFT} > 1$). The community weighted means (CWM) and standard deviations (CWSD) were obtained using the basal areas as weights.

Ap = *Acer pseudoplatanus,* Ca = *Corylus avellana,* Cm = *Crataegus monogyna,* Fe = *Fraxinus excelsior,* and Qr = *Quercus robur.* The colours of the different species in the first row of the Table are consistent with Figures 1 and 2.

| Trait | Ap | Ca | Cm | Fe | Qr | Others | CWM (± CWSD) |
|---|---|---|---|---|---|---|---|
| SLA ($m^2$ $kg_C^{-1}$) | - | 34.7 (± 36.1) | 62.8 (± 65.5) | - | 22.9 (± 23.9) | - | 25.1 (± 1.5) |
| $V_{c,max}$ ($\mu mol\ m^{-2}\ s^{-1}$) | 31.9 (± 16.1) | - | - | 39.7 (± 18.0) | 31.1 (± 18.8) | - | 32.6 (± 0.9) |
| PFT (if $N_{PFT} > 1$) | LH[1] | MH[1] | MH | MH | MH | MH | |
| **State variable** | | | | | | | **Total** |
| Density (-) | 532 | 67 | 24 | 84 | 35 | 73 | 815 |
| Relative density (%) | 65.3 | 8.2 | 2.9 | 10.3 | 4.3 | 9.0 | 100 |
| Basal area ($m^2$) | 31.59 | 0.48 | 0.24 | 5.96 | 11.87 | 0.57 | 50.71 |
| Relative basal area (%) | 62.3 | 0.9 | 0.5 | 11.8 | 23.4 | 1.1 | 100 |

[1]MH = Mid successional Hardwood trees, LH = Late successional Hardwood trees

**Table 2: List of varying processes included in the model ensembles in order to evaluate the model structural uncertainty as well as**
**their different possible configurations. Adapted from Shiklomanov et al. 2020.**

| Process | Description |
|---|---|
| **Crown model** | Choice of the crown representation in the canopy radiation model and in the turbulence scheme |
| Closed | Crowns are evenly spread throughout the patch area and cohorts are stacked on the top of each other |
| Finite | Cohorts have a finite radius and are stacked on the top of each other (Dietze et al. 2008) |
| **Radiative transfer model (RTM)** | Choice of the canopy radiation model |
| Two-stream | Two-stream approximation (Oleson et al. 2013; Sellers 1985) |
| Multi-scatter | Multiple-scatter approximation (Zhao and Qualls 2005) |
| **Trait plasticity** | Choice of including plant trait variation with the local environment |
| False | SLA and $V_{c,max}$ are constant |
| True | SLA and $V_{c,max}$ respectively increases and decreases with shading |
| **Plant functional diversity ($N_{PFT}$)** | Number of PFTs included in the simulation |
| 1 | All plant species are classified as mid-successional temperate deciduous trees |
| 2 | Plant species are mapped into two PFTs according to Table 1 classification |


**Table 3: List of allometries modified in this study, ED2.2 default and TLS-derived allometric coefficients (for one or multiple**
**simulated PFTs). The corresponding curves are plotted in Figure 2.**

| Allometry | Equation[1] | Parameter | ED2.2 default | | TLS $N_{PFT} = 1$ | $N_{PFT} = 2$ | |
|---|---|---|---|---|---|---|---|
| | | | MH[2] | LH[2] | MH | MH | LH |
| Height, h (m) | $h = h_{ref} + h_1 \cdot [1 - exp(DBH \cdot h_2)]$ | $h_{ref}$ | 1.3 | 1.3 | -3.2 | -3.2 | -2.8 |
| | | $h_1$ | 25.2 | 23.4 | 26.2 | 25.4 | 26.4 |
| | | $h_2$ | -0.05 | -0.054 | -0.074 | -0.074 | -0.07 |
| Aboveground woody biomass, $B_d$ (kg) | $B_d = B_{d1} \cdot DBH^{B_{d2}}$ | $B_{d1}$ | 0.16 | 0.24 | 0.37 | 0.67 | 0.23 |
| | | $B_{d2}$ | 2.46 | 2.25 | 2.29 | 2.13 | 2.42 |
| Crown area, CA (m²) | $CA = CA_1 \cdot DBH^{CA_2}$ | $CA_1$ | 2.49 | 2.49 | 0.6 | 1.4 | 0.3 |
| | | $CA_2$ | 0.81 | 0.81 | 1.15 | 0.95 | 1.33 |
| Leaf biomass, $B_l$ (kg) | $B_l = B_{l1} \cdot DBH^{B_{l2}}$ | $B_{l1}$ | 0.048 | 0.017 | 0.065 | 0.095 | 0.015 |
| | | $B_{l2}$ | 1.46 | 1.73 | 1.48 | 1.22 | 1.69 |

[1]DBH = Diameter at Breast Height (cm)
[2]MH = Mid successional Hardwood trees, LH = Late successional Hardwood trees


**Table 4: Description of the ED2.2 parameters varied in this stuy, their unit, and the definition of their prior used to evaluate the**
**model parameter uncertainty. "Source code name" is the name of the parameter as it appears in the ED2.2 source code. When trait**
**plasticity is enabled, both SLA and $V_{c,max}$ may change over time and for different cohorts of the same PFT.**

| Parameter name | Description | Unit | Prior | | | Source code name |
|---|---|---|---|---|---|---|
| | | | Function[1] | a[2] | b[2] | |
| Water cond. | Soil-plant hydraulic conductance | $m^2 (kg_{C,root})^{-1} yr^{-1}$ | lnorm | -10.8 | 3.5 | water_conductance |
| Growth resp. | Fraction of assimilation lost to growth respiration | Unitess (0-1) | beta | 4.06 | 7.2 | growth_resp_factor |
| Mort. C bal. | C balance ratio at which mortality rapidly increases | Unitless | gamma | 1.47 | 0.058 | mort2 |
| $V_{c,max}$ | Maximum rate of $CO_2$ carboxylation at 15°C (baseline) | $\mu mol_C\ m^{-2}\ s^{-1}$ | weibull | 1.7 | 80 | Vm0 |
| Leaf resp. | Leaf dark respiration at 15°C | $\mu mol_C\ m^{-2}\ s^{-1}$ | gamma | 1.5 | 0.4 | Rd0 |
| Root:leaf | Ratio of fine root to leaf biomass | Unitless | lnorm | 0.21 | 0.6 | q |
| SLA | Specific leaf area (baseline) | $m^2 (kg_{C,leaf})^{-1}$ | gamma | 5.13 | 0.23 | SLA |
| Clumping | Canopy clumping factor | Unitless (0-1) | beta | 3 | 1.5 | clumping_factor |
| Quant. eff. | Fraction of absorbed light used for $CO_2$ fixation | $mol\ CO_2 (mol\ photon)^{-1}$ | weibull | 3.32 | 0.08 | quantum_efficiency |
| Refl. (VIS) | Leaf reflectance in the visible range (400-700 nm) | Unitless (0-1) | beta | 10.1 | 157 | leaf_reflect_vis |
| Refl. (NIR) | Leaf reflectance in the NIR[3] range (700-2500 nm) | Unitless (0-1) | beta | 35 | 56 | leaf_reflect_nir |
| Stomatal slope | Slope between leaf assimilation and stomatal conductance (Leuning) | Unitless | lnorm | 2.3 | 1 | stomatal_slope |
| Min. height | Minimum height for plant reproduction | m | gamma | 1.5 | 0.2 | repro_min_h |

[1]lnorm = log-normal distribution
[2]The values a and b define the parameters of the prior distributions (LeBauer et al., 2013).
[3]NIR = near-infrared


**Table 5: Summary of the model configurations used in this study and the underlying model settings.**

| | | Configuration name | | |
|---|---|---|---|---|
| | | NBG | Census | TLS |
| Settings | Initial conditions | Near-bare ground | Inventory | TLS |
| | Allometric parameters | Unconstrained | Unconstrained | TLS-constrained |
| | Run length (years) | 100 | 5 | 5 |
| | Crown model | Closed or finite | Closed or finite | Finite |
| | RTM | Two-stream or multi-scatter | | |
| | Trait plasticity | True or false | | |
| | $N_{PFT}$ | 1 or 2 | | |
| | Ensemble size | 500 | | |


**Table 6:** Summary of most important states and fluxes in all three model configurations and how they compare with observational datasets, including flux tower data of ecosystem respiration and net ecosystem productivity. Those numbers take into account the full five years of simulation for the prescribed model configurations (Census and TLS), and the last five years of simulation for the near bare-ground conditions (NBG), and the two years of eddy covariance observational data. For the observations of LAI in the leaf-on season, we provide a range of variation.

LAI = Leaf Area Index, AGB = Aboveground Biomass, GPP = Gross Primary Production, NEP = Net Ecosystem Productivity, PAR = Photosynthetically Active Radiation

| | Units | Configuration | | | | | Observations |
|---|---|---|---|---|---|---|---|
| | | NBG | | Census | | TLS | |
| | | Closed canopies | Finite crowns | Closed canopies | Finite crowns | Finite crowns | |
| AGB | $kg_C\ m^{-2}$ | 11.9 ± 7.4 | 10.8 ± 6.8 | 16.4 ± 5.3 | 17.1 ± 4.7 | 24.5 ± 2.5 | - |
| **Leaf-on only period (May to October)** | | | | | | | |
| LAI | $m^2\ m^{-2}$ | 3.83 ± 1.94 | 4.72 ± 3.67 | 4.71 ± 1.28 | 5.75 ± 2.74 | 3.79 ± 0.50 | 3.6 - 4.1[2] |
| PAR reaching the ground | $\mu mol\ m^{-2}\ s^{-1}$ | 78.6 ± 93.2 | 90.9 ± 95.4 | 44.8 ± 34.7 | 58.2 ± 35.3 | 98.2 ± 36.0 | - |
| GPP | $\mu mol\ m^{-2}\ s^{-1}$ | 9.55 ± 4.34 | 9.81 ± 4.70 | 10.94 ± 2.91 | 11.83 ± 2.95 | 9.86 ± 2.89 | 9.8 ± 3.4[1] |
| Ecosystem respiration | $\mu mol\ m^{-2}\ s^{-1}$ | 6.92 ± 3.13 | 7.03 ± 3.43 | 7.03 ± 1.82 | 7.32 ± 1.80 | 6.07 ± 1.81 | 7.2 ± 1.3[1] |
| NEP | $\mu mol\ m^{-2}\ s^{-1}$ | 2.63 ± 1.46 | 2.78 ± 1.49 | 3.91 ± 1.74 | 4.51 ± 1.92 | 3.79 ± 1.67 | 2.6 ± 2.5[1] |
| **All year round** | | | | | | | |
| GPP | $\mu mol\ m^{-2}\ s^{-1}$ | 6.04 ± 2.77 | 6.26 ± 3.02 | 6.88 ± 1.84 | 7.46 ± 1.87 | 6.24 ± 1.85 | 5.5 ± 4.7[1] |
| Ecosystem respiration | $\mu mol\ m^{-2}\ s^{-1}$ | 4.51 ± 2.04 | 4.64 ± 2.24 | 4.56 ± 1.16 | 4.78 ± 1.15 | 3.98 ± 1.17 | 5.3 ± 2.1[1] |
| NEP | $\mu mol\ m^{-2}\ s^{-1}$ | 1.53 ± 0.86 | 1.63 ± 0.89 | 2.32 ± 1.05 | 2.68 ± 0.42 | 2.26 ± 1.02 | 0.3 ± 2.9[1] |

[1]Reference: Thomas et al. (2011) and Fenn et al. (2015).
[2]Reference: Roberts et al. (1999)

**Figures**

928

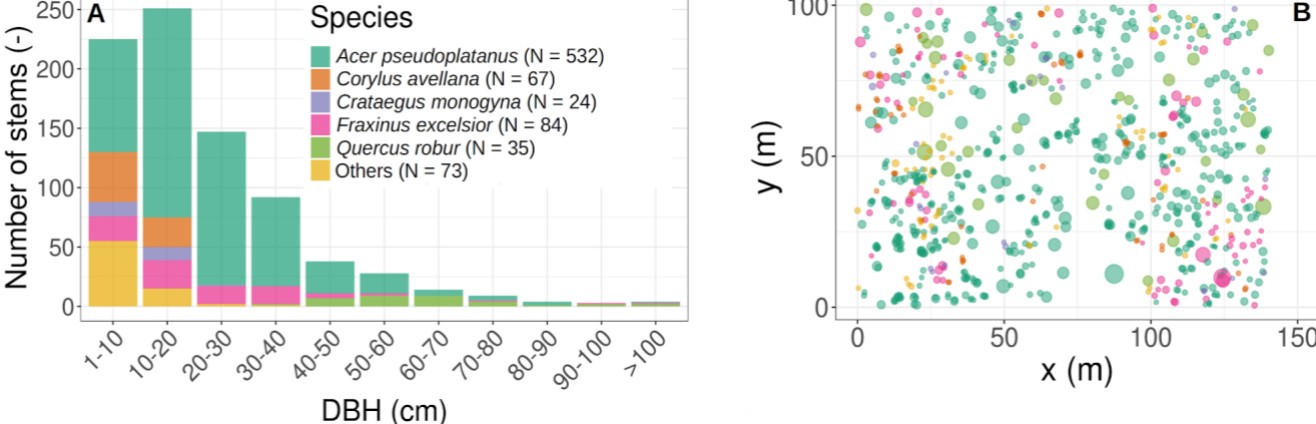

929

**Figure 1: Initial conditions in terms of tree size distribution and species composition (A), horizontal position, basal area (the size of the circles in panel B is proportional to the individual basal area), and species composition (B). The species colour legend applies to both panels and is kept the same for Figure 2 and Table 1. In the simulations, all trees were classified into either a single or multiple plant functional types according to the species-PFT of Table 1.**

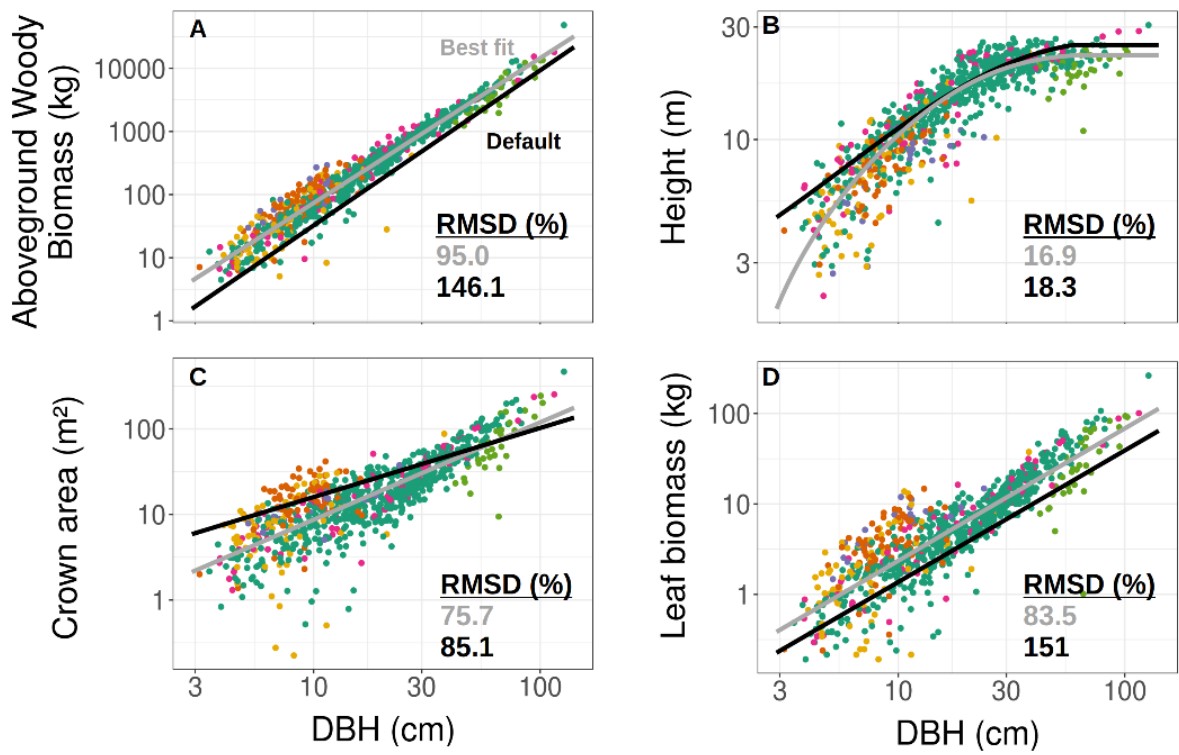

**Figure 2: TLS-derived (grey, considering all tree species belonging to a single PFT) and model default (black, mid successional**
**hardwood trees in ED2) allometries for the aboveground woody biomass (A), tree height (B), crown area (C), and leaf biomass (D).**
**The data to which the TLS allometries were fitted (coloured points corresponding to the tree species detailed in Figure 1) were**
**obtained using TLS. Coefficients used to plot the best fit and default allometries can be found in Table 3.**

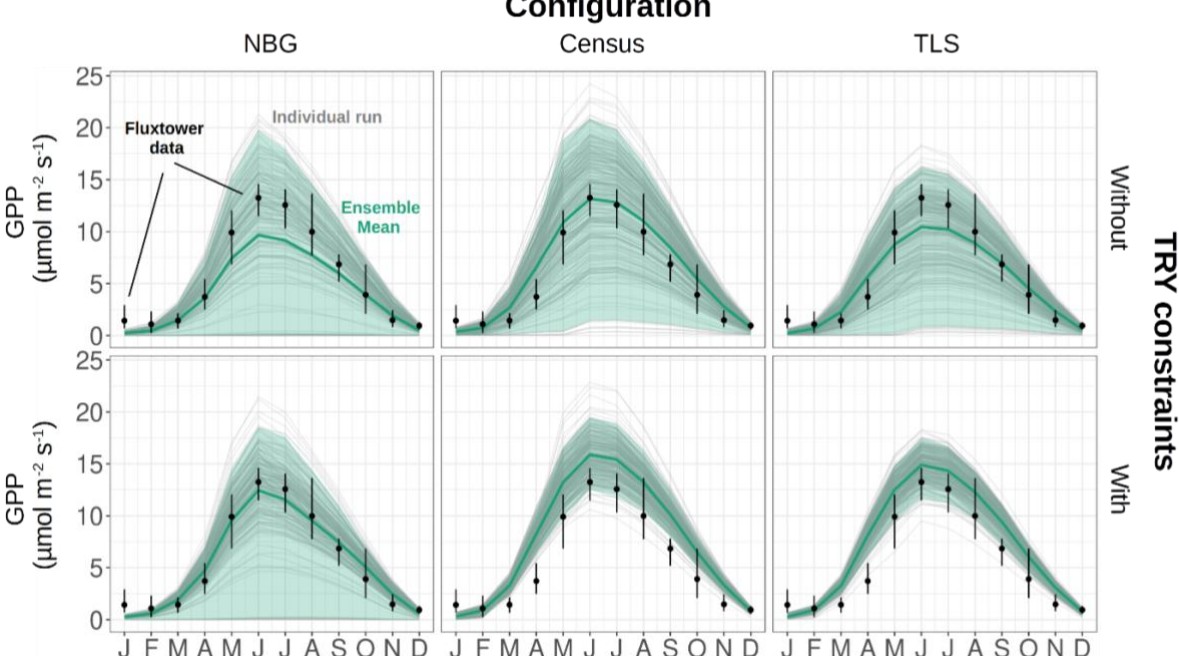

Figure 3: Seasonal cycle of the ecosystem GPP, as observed by eddy-covariance data (black dots) or as simulated by ED2.2 for
multiple model configurations (columns) and with or without TRY constraints on SLA and $V_{c,max}$ (rows). The green thick lines are
the ensemble means while the shaded envelopes encompass 95% of the ensemble members. The individual ensemble members are
also plotted as thin grey lines. The vertical error bars for the flux tower data represent the 95% confidence interval of the monthly
GPP. The settings of the model configurations are detailed in Table 5.

# Configuration

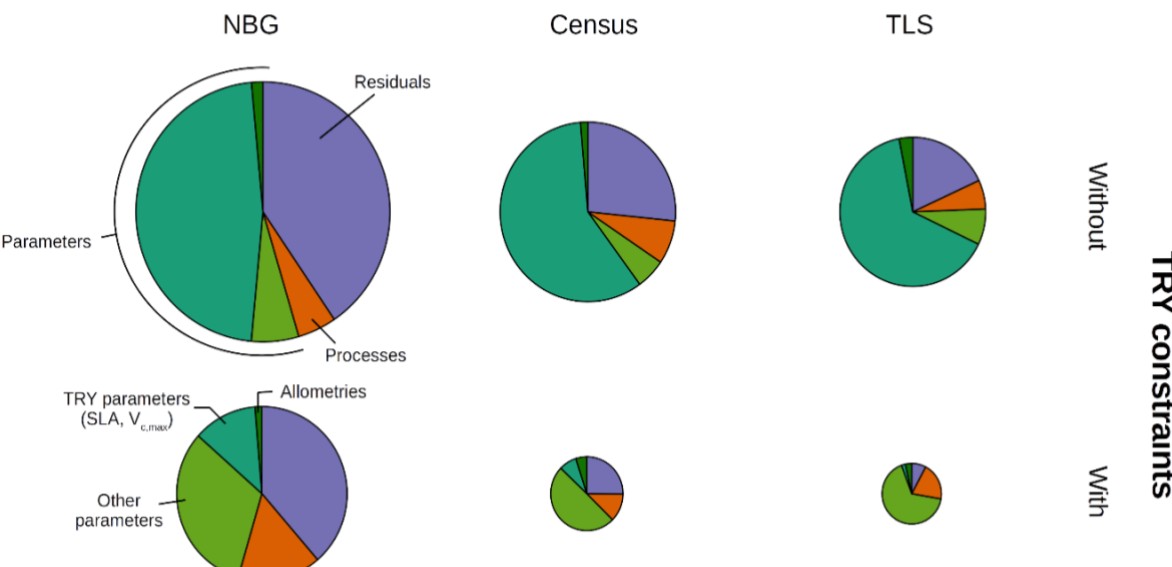

**Figure 4: Decomposition of the simulated GPP variance into process (orange), parameter (green), and residual (mauve) uncertainty**
**for multiple model configurations (columns) and with or without TRY constraints on SLA and $V_{c,max}$ (rows). The parameter**
**uncertainty was further decomposed into the contribution of the allometric, TRY-constrainable (SLA and $V_{c,max}$), and other**
**parameters (shades of green). The radii of the pie charts are proportional to the total variance of the ecosystem GPP in each**
**configuration for the month of June (maximum GPP). The settings of the model configurations are detailed in Table 5.**

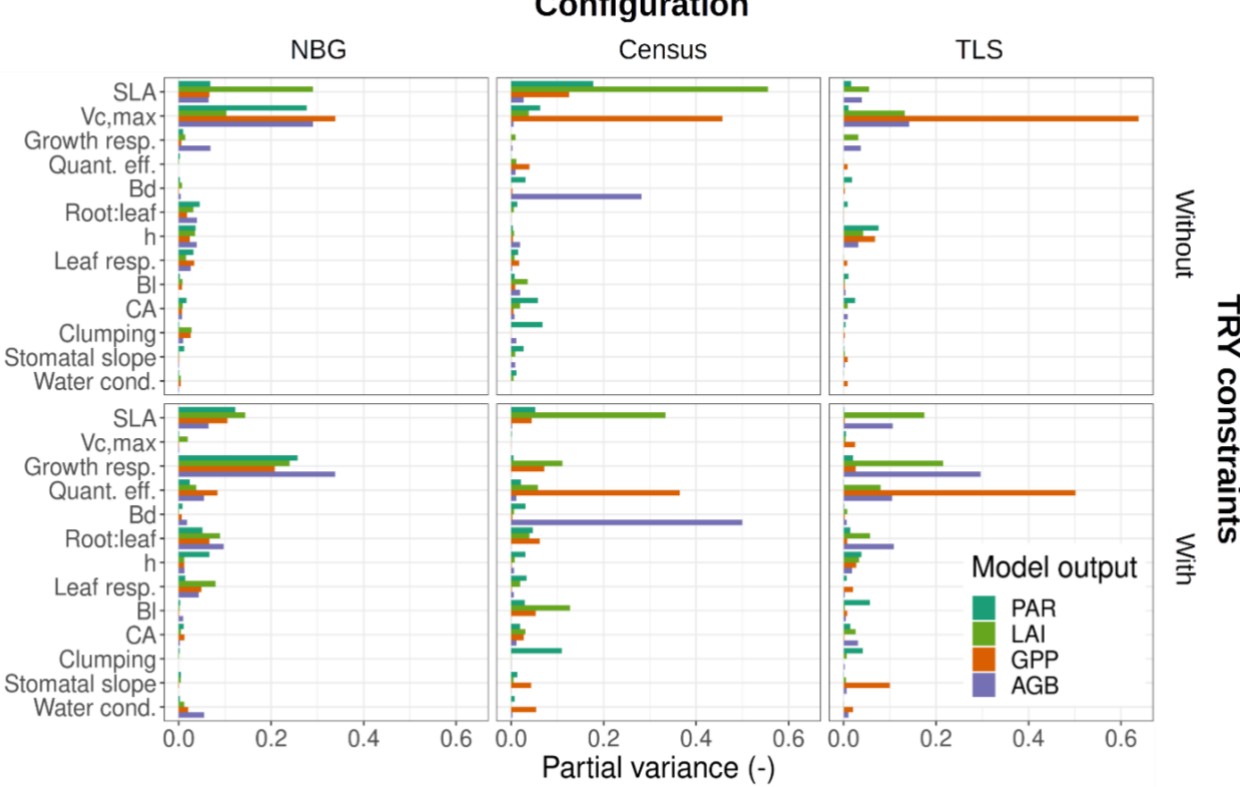

**Figure 5: Contribution of individual or allometric parameters (Bd, Bl, CA and height include all parameters for the respective allometries, see Table 2) to the predicted uncertainty in ED2.2 of multiple state variables (PAR = photosynthetically active radiation reaching the ground, LAI = leaf-on ecosystem leaf area index, AGB = final ecosystem aboveground biomass, GPP = leaf-on ecosystem gross primary production) for multiple model configuration (columns) and with or without TRY constraints on SLA and $V_{c,max}$ (rows). Only those parameters that contributed at least once to 5% or more of the total variance were included in the panels. Parameter description and distributions are given in Table 4. The settings of the model configurations are detailed in Table 5.**



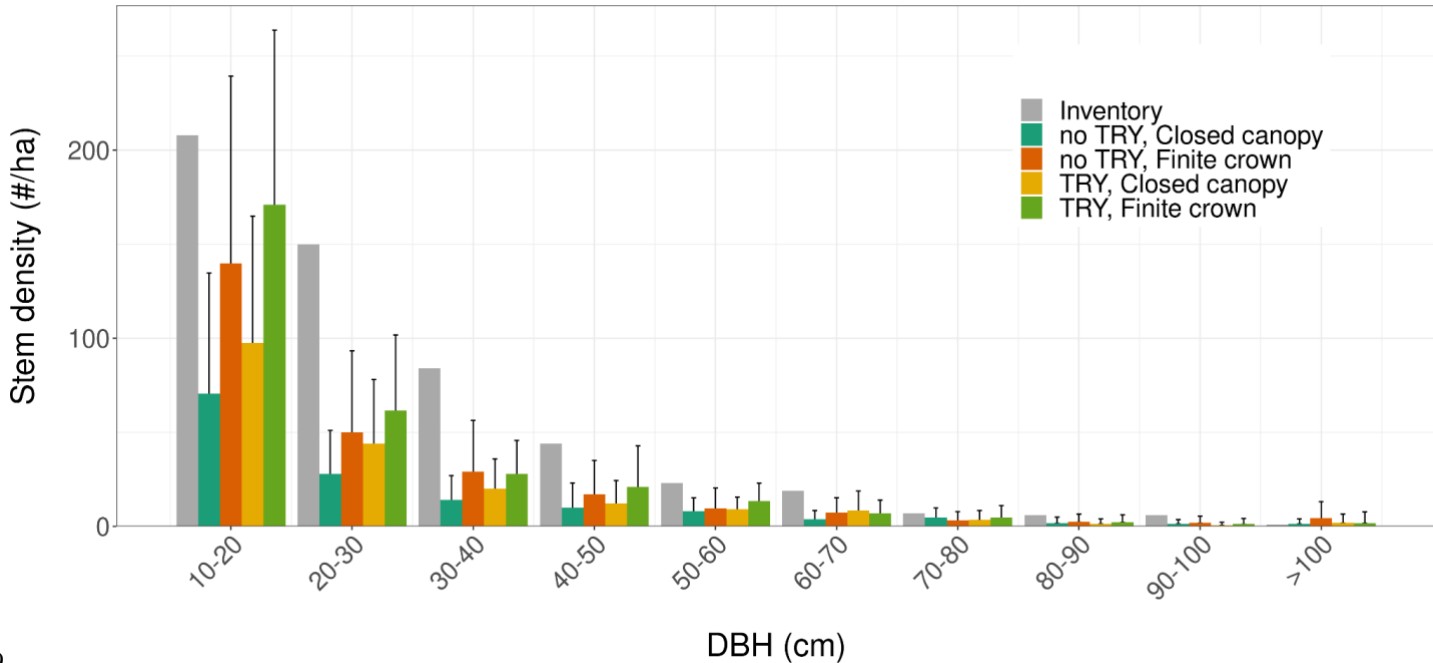

Figure 6: Tree size distribution for multiple model configurations starting from near bare-ground conditions after 100 years of simulations (coloured bars), and how they compare to the field inventory (grey). The histograms and the vertical error bars represent the means ± one standard deviation of the ensemble member runs. Only runs that generated vegetation were kept for plotting this figure.

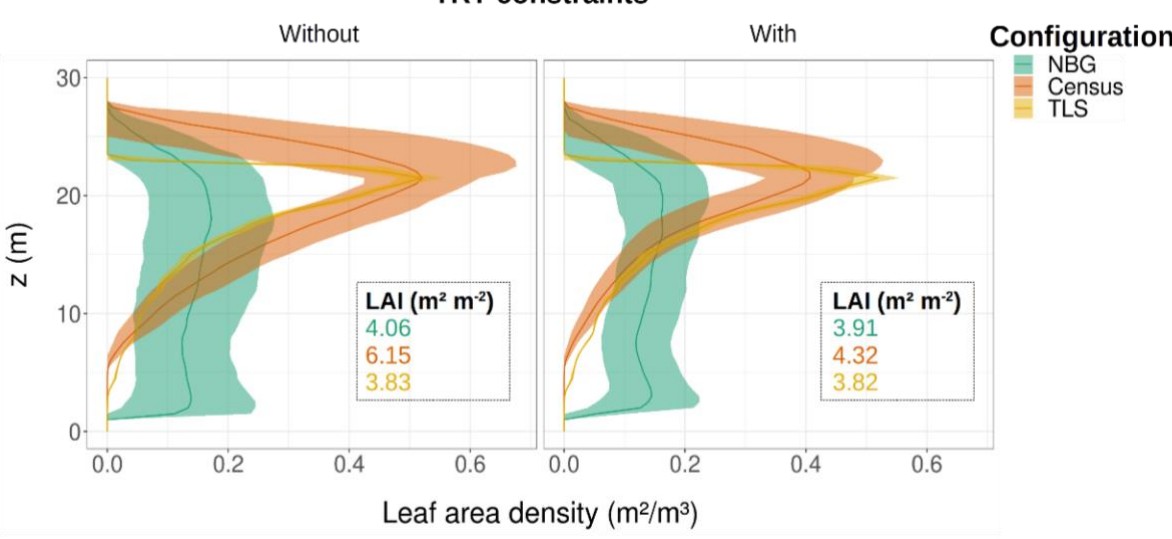

Figure 7: Ecosystem average of the leaf area density vertical distribution for the month of June for different model configurations (colourd lines and envelopes) without (left) and with (right) TRY constraints on SLA and $V_{C,max}$. The envelopes encompass the mean ± one standard deviation of the ensemble member runs. Only runs that generated vegetation were kept for plotting the NBG envelopes. The settings of the model configurations are detailed in Table 5.