# Peer review of "Using terrestrial laser scanning to constrain forest ecosystem structure and functions in the Ecosystem Demography model (ED2.2)"

_Geoscientific Model Development, 2021_

## Author Comment (AC1)

**Response to anonymous referee # 1**

The concept for this study is strong, using TLS to constrain forest structure and function in the ED2.2 model, follows a decade and a half's work on using remote sensing to constrain predictions made by ecosystem models (in reducing process and initialization errors). While this idea is worth publishing, the execution is not clear, the structure of the study needs improvement, and the actual constraining of the ED2.2 model is adequately done. Concerning this last point. Essentially you want to know how well your TLS-constrained ED2.2 simulations has fared compared to Ground-based-initialized ED2.2 simulations and compared to bare-ground simulations. To assess the improvement you need to compare all 3 of these simulations to observed data (like GPP, plot basal area changes, and/or growth and mortality dynamics). You need to do this for both TLS-structure and TLS allometric improvements.

**We first would like to thank Reviewer #1 for their positive review of our work. We were pleased to see that he/she shares our opinion that TLS has strong potential to successfully constrain forest models and that such an idea is worth publishing In Geoscientific Model Development. After reading both reviews, we decided to reshape our study in order to improve the structure of the manuscript and enhance the execution of the research idea. The concepts, data, and methods will remain essentially the same but will be reshaped, used and presented in a different manner.**

**As highlighted by both reviewers, what we are interested in at the end of the day is (i) the sensitivity of the ED2 model to parameters and processes that can be constrained/defined via TLS-derived observations and (ii) how model simulations constrained to TLS data compared to ground-based initialized simulations, bare-ground model runs and independent observations. In order to investigate this, we propose to replace the current**

set of analyses by an overall global sensitivity and variance decomposition analysis applied to different model configurations (default and TLS-informed). This new, large analysis will encompass the previous three analyses and extend them. For each model configuration, we will run large ensembles of the Ecosystem Demography model and decompose the total uncertainty into its components (initial conditions, model structure and model parameters). This will allow us to answer three types of research questions that were somehow already present in the previous version of the manuscript but only partly identified and answered. Those research questions are:

(i) What is the total model uncertainty of the ED2 model for the specific site that we try to mimic in silico? What are the contributions of the different sources of uncertainty? And how do they change along the simulation?

(ii) Is the total model uncertainty reduced when constraining the model with TLS observations? Do the primary sources of uncertainty remain the same?

(iii) Does the use of TLS data improve the model performance?

Model ensembles will be started from TLS-derived inventories (TLS-informed configuration) or from near bareground conditions/field inventories (default configuration). The importance of the model structure will be tested in the different configurations by changing the crown representation (finite or infinite) and the number of simulated PFTs (see below for details)  while the impact of model parameters will be quantified by changing a larger set of parameters than just SLA and $V_{cmax}$ as suggested by the second reviewer. We propose to include all parameters that were shown to drive a significant fraction of the model uncertainty in previous studies (see e.g. Dietze et al., 2014 and Shiklomanov et al., 2019) and not only the ones for which we have trait data available. In the TLS configuration, the allometric parameters will be constrained to TLS

**data while the prior distributions of those parameters will remain uninformed in the default configuration.**

**Once achieved, those ensembles will directly serve to quantify the total model uncertainty and partition it into its components (initial conditions, model structure, and parameters). A simple comparison of TLS-informed *vs* default model configurations will allow us to estimate the benefits of TLS to reduce the total model uncertainty and how its drivers change when some processes and parameters are constrained by TLS data. Finally model outputs will be compared to the datasets previously used to calibrate some of the model configurations (eddy covariance fluxes, LAI and light observations). Doing so, we will now be able to compare the performance of the different configurations as well as some of their subsets (e.g. TLS-structure and TLS allometries).**

There are many more comments in the text attachment below.

**We thank reviewer #1 for this detailed review, which is very useful to correct all minor points. As we expect the manuscript to change significantly (as explained above), we do not provide here a detailed response for every single comment. Instead, below we only repeated and answered the comments that will remain relevant for the new version of the manuscript and the new analyses. Should some of those comments that we left open appear to be relevant during our revision, we will of course take the reviewer's suggestion into account.**

Detailed comments and response

L25: "… at a temperate forest site" As it is a single site, isn't it meaningful to say what it is?

**We will name the site (Wytham Woods Forest, UK) starting from the abstract**

L27: "… productivity …" gross primary productivity? net primary productivity? leaf area production?

**The global sensitivity and variance decomposition analyses will be performed on several model outputs, including gross and net primary productivity, as well as the total leaf area.**

L29: "the imposed openness" do you mean the gap fraction here?

**By "imposed openness" we meant "the representation of the canopy (infinite or finite crown areas)". That will be clarified in the new version.**

L35-39: "We conclude … and reduce their overall uncertainties" Have you validated you new TLS-configured ED2.2 runs? Do you have tested reductions in uncertainty?

**As explained above, validating the TLS-informed runs with independent datasets (including eddy-covariance data, basal area changes, etc.) is one of the objectives of the new analysis and will be presented in the new version of the manuscript.**

L165-166: "All trees inventoried in Wytham Woods were classified as mid-successional temperate deciduous trees" Are you sure. Sycamore can be considered a shade-tolerant tree (late successional), maybe hazel. You need to go through each and every species to see which are early, mid and late successional species. Identifying them all as mid-successional needs a good explanation.

**As the manuscript was initially seen as a simple sensitivity analysis of the ED2 model to the parameters and processes that could be constrained to TLS data, we wanted to simplify the analysis as much as possible and hence decided to simulate a single PFT. Incorporating more than one PFT would have made the whole analysis way more complicated to present, as well as for rendering the outputs to the readers. Yet, with the new global analysis that we foresee, it will be possible to take this complexity into account by simulating one or several PFTs. The tree species mapping to the model PFT will be achieved using trait data of SLA and wood density, the allometric relationships derived from TLS, as well as previous classifications from the literature.**

L191: "i) near-bare ground initial conditions (i.e. seedlings only" Is this with Mid-Successional Hardwoods seedlings only?

**In the previous runs, it was indeed with Mid-Successional Hardwoods only. In the new runs, it will be Mid-Successional Hardwood trees or a combination of several PFTs (see comment above).**

L191-192: "the DBH distribution available through the field inventory" Why not just call this the 'ground-inventory initialization'? Up to you.

**That is what we will call it in the revision.**

L193-194: "that were allowed to fuse along the simulation" Unclear. Delete?

**This will be deleted.**

L194-195: "Simulations were run … dataset (Viovy 2018)" We need more information on the initializations. How long was the bare-ground simulation run for? How did you cycle the years of the met data? Which PFTs did you use? What soil data were used at the site? What is the resolution of your met data? What are the structural attributes of the ground vs TLS initial forests...i.e. Basal Area, DBH, LAI, TLS occlusion etc.

**Those pieces of information will be provided in the new version of the manuscript.**

L214-216: Can you have results for this analysis and comparisons to ground-inventories?

**In the revised version, all near bare-ground runs will be compared to ground inventories as an independent validation.**

L231: "Parameter data assimilation and model equifinality (Analysis III)" Rather than equifinality, this method appears to be about parameter optimization. Perhaps I am wrong.

**Yes, you are correct. Yet analysis III will no longer be included in the new version of the manuscript.**

L261-264: I can't find these numbers in the figures

**Those numbers will be added in the new version of Figure 3.**

L265-271: These results are not clear. It is assumed you are describing Figure 3, but numbers comparing TLS and ED2 are not clear.

**Figure description will be improved in the new version.**

L273-275: TLS and ground differences are important here and should not be in the supplement. Valuing the use of TLS in ED2.2 needs a rigorous comparison to ground-inventory data and initializations.

**We will add a proper comparison of the TLS and ground inventory in the main text in the new version.**

L277-278: Figure 6 has the IC- parameter optimization results!

**This issue will be corrected.**

L287: Start off making reference to the Figure you are about to explain

**We will make sure we properly start off making references to the appropriate figures in the new version of the manuscript.**

L304: It is still unclear why you are doing this step, and why you are not including the new optimized results in the sensitivity analysis (Figure 5). Also it seems counter-intuitive that IC-Default Vcmax value of 47.3 compared to 17.5 would not largely affect GPP!

**The $V_{cmax}$ definitely influenced the modelled GPP but the effect of the parameter was compensated by the reduction in SLA resulting from the optimization, which eventually led to similar GPP. In the new analyses we will no more perform optimizations but rather run a global sensitivity analysis and compare the (sub-)ensembles with the (now) independent eddy-covariance datasets.**

Figure 7: Confusing figure. What this tells me is that the TLS and FC initial conditions are not that close, especially with the results in SLA. How useful is this optimization, and why do this

if you are not going to use it in the sensitivity (Figure 5), or in ultimately comparing the improvements of TLS vs ground-inventory initializations against carbon fluxes?

**The optimization analysis will not be repeated in the new version of the manuscript.**

Figure 7 (bis) What does this mean?/What is this dot?

**The dots were the data from the TRY database. Yet, this figure is likely to disappear in the new version.**

L355: There is no Table 5

**We are sorry that Table 5 was missing in the manuscript. It disappeared somewhere in the submission process. We will make sure those data are available in the new version.**

L359-365: It may be expected that better site level descriptions of BLeaf allometry will affect the LAI and GPP, and that better descriptions of Bdead will affect Woody biomass. Rather than sensitivity, what about overall improvement to LAI, GPP, Biomass??

**That is going to be the goal of the new analysis according to your and reviewer #2' suggestions.**

L379-380: Have you done model calibration effectively using TLS?

**That is indeed what we did in the previous version of the study but we will not do this step in the next version as explained above.**

L383-384: Not sure you have done showing the effective improvement in model performance.

**We hope that the new analysis will allow us to conclude about the effectiveness of using TLS data to constrain the model uncertainty and improve model performance. Based on the model sensitivity and the results presented in this first version, we expect a strong reduction in uncertainty and a significant improvement of model performance thanks to the TLS data.**

L397-398: From your results, this 'equifinality' issue could be demonstrated without TLS, just by optimizing using bareground and ground-inventory initializations. Furthermore, the usefulness or added value of this equifinality/optimization excercise is still not clear.

**The optimization exercise will no longer be present in the new version of the manuscript and equifinality will definitely emerge from the large ensemble runs (especially in the default configuration).**

L403-L404: You could use recurring forest inventories collected at your site as an additional calibration dataset

**We did not use successive inventories from Wytham Woods in the first version of the manuscript but these data will be added as an additional validation dataset in the second version if we can get access to it.**

---

## Author Comment (AC2)

**Response to anonymous referee # 2**

The manuscript by Meunier et al., uses TLS data to inform coefficients of an ecosystem model's allometric equations and initial conditions, quantifies its impact, as well as testing influence of TLS information on model calibration. While the study is well thought out and generally well-written and visualized, there are some issues with both modelling and calibration protocols (in terms of both technicality and clarity). Also the manuscript remains somewhat inconclusive about the superiority of TLS-informed model predictions, or at least if that wasn't the case the manuscript needs to be revised to clearly present it as such. It's a pity Table 5 was not available for the review process. Overall, I think the study would be of interest for the community and worth publishing, however, I would strongly recommend tackling the technical issues raised by both reviewers. Line numbers below refer to the author's preprint.

**First, we would like to thank reviewer #2 for their thorough assessment of our manuscript. We believe that the comments that were raised are fair and will contribute to improve the overall quality of the study. Their suggestions fall in line with the ones of the first reviewer and we agree with both of you that despite being interesting and worth publishing (thanks for pointing that out!), the results were probably not presented in the clearest way. We therefore propose to reshape the manuscript around one central analysis, which will replace and complete the previous three. The concepts, data, and model will remain essentially the same but will be analysed and presented in a somewhat different manner. The new analysis will assess the global sensitivity of the model and evaluate its performance when the simulations are constrained or not by TLS data. More precisely, we will run large ensembles under different model configurations (default and TLS-informed),**

and partition the overall uncertainty into its components (initial conditions, model structure, and model parameters).

Model runs will be initialized from near bare-ground or from field inventory (default) or directly from TLS-derived size distribution (TLS). For the model structure, we will test the impact of the crown representation (finite or infinite) as well as of simulating more than a single PFT. Additionally, we will increase the number of parameters to be tested to include those that were shown to be significantly contributing to the overall model uncertainty in previous ED2 studies. This global sensitivity and variance decomposition analyses will allow us to investigate the following research questions that were somehow present in the previous version of the manuscript but not clearly identified nor fully answered:

(i) What is the total model uncertainty? What are the contributions of the different sources of uncertainty? And how do they change along the simulation?

(ii) Is the total model uncertainty reduced when constraining the model to TLS observations? Do the primary sources of uncertainty remain the same?

(iii) Does the use of TLS data improve model performance?

To answer the last question, we will use the independent datasets that were previously used for model calibration (eddy-covariance fluxes, LAI and light observations) as well as recurring forest inventories as validation datasets and compare how the ensembles reproduce those. Most of those data were included in Table 5 and we are sorry that it somehow disappeared during the submission process. We are very confident that such a reshape of both the study and the manuscript will significantly improve the technicality and clarity of the protocole and clearly shows the positive impact of the use of TLS data on model uncertainty and performance.

**We also thank reviewer #2 for the detailed review, which is very useful to correct all minor points. However, as we expect the manuscript to change significantly because of the reshape of the analyses, some of the comments will no longer be relevant. For that reason, below we only answered the comments that will remain relevant for the new version of the manuscript and the new analysis. Should some of those comments that we left open appear to be relevant during our revision, we will of course take the reviewer's suggestion into account.**

Title: As mentioned in the general comment above, the manuscript is rather inconclusive about the reliability of TLS informed model predictions. Even the abstract reports only the sensitivity of the results to model configuration and TLS information. Hence, it feels as if the title would reflect the study more closely if it was revised to something along the lines of "Sensitivity of ED2.2 forest ecosystem simulations to TLS informed/constrained structure and functions" (as also presented by the authors on L95 and L195).

**At the moment, we agree that we mainly tested the sensitivity of the model simulations to TLS-informed structure and functions and therefore a title such as the one suggested would be probably more appropriate. Yet, we think (based on the results presented in the first version) that the new version of the model analysis will allow us to be more conclusive about the reliability of the TLS-informed simulations, and therefore hope that we will be able to keep this (or a similar) manuscript title.**

L28: "imposed openness" do you mean the FC configuration here? If yes, please revise to explicitly say "model configuration that imposes finite canopy radius dramatically influenced..."

**We indeed meant the crown representation here (finite or infinite) and this will be made clear in the next version of the manuscript.**

L33-34: After reading the manuscript, I wasn't quite left with a conclusion about the most adequate model structure. If you identified it, why not say it in the abstract explicitly.

**Indeed, the previous set of analyses did not allow to identify the most adequate model structure because we lacked independent datasets (eddy covariance fluxes were used to calibrate the model, and LAI observations were derived from the same TLS datasets which served to parameterize the model). Reshaping the analysis as we suggest above will allow us to reach such a conclusion: we will now be able to see the benefits of constraining some of the model parameters and processes to TLS data on the model uncertainty and performance.**

L81: Somewhere around this paragraph I would have expected a brief introduction about other (e.g. airborne) lidar studies with TBMs as well. Especially given that studies exist directly with the ED model, Hurtt et al. 2004 (https://doi.org/10.1890/02-5317), 2019 (https://doi.org/10.1088/1748-9326/ab0bbe), Thomas et al., 2008 (https://doi.org/10.5589/m08-036). I think this could benefit the discussion as well, e.g. what did the authors build upon the previous lidar-ED2 studies? or they can draw parallels to this study.

**While very relevant, those studies were unknown to us. They will definitely be included in the introduction and discussion of the new version.**

L136-140: Appreciated the length authors went with extracting the data. However, this paragraph would benefit from further information on the overall quality of the data: what the frequency of the data is (daily, sub-daily?), how it was filtered, QA/QC'd, how the GPP was derived, what the accuracy of data retrieval from Plot digitizer software is, if there are known issues with the time series that could affect the calibration and so on.

**In the future version of the manuscript, we will provide these pieces of information.**

L152: I'd like to point out that the authors themselves avoid using the word "validation" here, which again reinforces my comment about inconclusiveness. In case you decide to strengthen the paper's conclusions, at least consider the word "assessment" here.

**The trait data will now serve to inform the parameter distribution before the global sensitivity analyses. Model performance will be assessed with other independent datasets (including the eddy covariance fluxes) that were previously used to calibrate some of the model configurations. In that sense, we will use those observations as validation datasets and will be truly able to compare model performance with and without TLS constraints.**

L164: Agreed with the other reviewer. Why was all classified as mid-successional pft in ED? I agree that each species, at least the five on Figure 1, needs reasoning as to which PFT they were mapped to and why. Please also provide citations for mappings when possible e.g. see supplementary on https://onlinelibrary.wiley.com/doi/full/10.1111/j.1365-2486.2011.02477.x where Acer is LH, Quercus is NMH. Admittedly, using multiple PFTs would complicate the reporting as authors are currently only concerned with a single set of allometric parameters, but worth exploring. Also, even if the authors decide to continue with a single PFT after revisions, they should emphasize already here that this is an over-simplification which could help prevent misuse by others referring to this study in the future.

**The first version of our study mainly aimed at assessing the sensitivity of the ED2 model to parameters and processes that could be constrained by TLS data. At this stage we wanted to simplify the model complexity as much as possible to render a clear and simple message. In that sense, increasing the number of PFTs, while important to simulate the complexity and diversity of acquisition strategies of the tree species, would have in our opinion made our analyses and conclusions fuzzier and more difficult to grasp. We**

**propose to overcome the issue by incorporating the number of modelled PFTs in the model structure uncertainty and doing so to quantify the uncertainty that is associated with this simplification. The tree species mapping to the model PFT will be achieved using trait data of SLA and wood density, the allometric relationships derived from TLS, as well as previous classifications from the literature, as suggested by the reviewer.**

L192: Agreed with the other reviewer. Please provide more details or point to the initialization/settings files of ED2.2 specifically if you have deposited them to the repository cited at the end (you could have a supplementary table telling which initialization/settings files went with which experiment or populate the readme file on the repository) .

**In the next revision, we will add the missing information in the main body and include the settings files of the model in the Zenodo repository.**

Figure 2 is great, but I'd call Analysis III: Bayesian calibration instead of data assimilation to be more precise, or at least continue using "parameter data assimilation". Also for analysis I, did you use TLS to inform structure directly? Looking at Table 4 it's only allometries. Allometries in return affect the structure but if I saw only allometries in that box, it would have helped me follow the study better.

**Indeed it should have been called "parameter data assimilation" or "Bayesian calibration". This analysis will no longer appear in the revised version of the manuscript and Figure 2 will therefore be updated.**

L202-203: What do you mean by "to assess the relative importance of TLS we compared it to field observations"? Does this exercise result in Fig S1 and S2? Isn't it then better to call this ground-truthing or validation of TLS? Please clarify.

**Indeed this corresponds to Figure S1 and S2. We will call this "ground-truthing of TLS" in the revisions.**

L207: Could you already explain here if 100 years spinup is enough, especially considering that the actual age of the forest is much older? I know of other models running much longer spinups (e.g. 500 years), please motivate the reader if 100 years is appropriate.

**The 100 year-long run does not correspond to a true model spinup but rather to the approximate age of the forest (it is the last time since large-scale disturbance occurred to Wytham Woods to the best of our knowledge). So if the model was perfect, the size distribution as simulated after 100 years should resemble the actual observation. We will make this clear in the revision**

 L214: Looking at Table 4, how about NBG-infinitely wide-TLS setup? See comment below regarding having another control for impact of TLS informed allometries.

**We did not include the NBG-infinitely wide-TLS setup because TLS data allowed us to constrain the crown area allometry. Yet, in our new analysis the NBG-infinitely wide TLS will be necessarily included in the ensembles.**

L221:  Why not explicitly state in what order these changes and combinations were introduced as this might also help following the incremental effect discussion. Listing configurations for 16 runs is not that much, could be also in the supplementary.

L226-228: Agreed with the other reviewer on quantification of indirect effects. I think listing all the configurations for the mentioned 16 runs will help. I assume authors performed a factorial design here but it is not clear which combinations went with which.

**For both previous comments: the new global sensitivity analysis will not use this factorial design anymore so we will get rid of those issues while keeping the main results of the sensitivity analysis.**

L231: "parameter optimization by Bayesian data assimilation" -> Authors could consider using "Bayesian parameter data assimilation" here as well to be more clear. Or better yet, "Bayesian calibration of model parameters".

**Indeed Bayesian calibration or Bayesian parameter data assimilation sound better.**

L235: Looking at Table 4, it feels like there needs to be another intermediate setup: inventory-finite radius-default, is there a particular reason why authors omitted this configuration? Also this sentence on L231-L233 suggests this configuration was included but Table 4 does not mention this configuration: "The model configurations included a default model version (default allometric parameters, infinite crown area), and a finite crown representation (default allometric parameters, finite crown radius), *which were both initialized with field inventory data*" I believe, according to this sentence, Table 4 second to last column should read "inventory" for initial conditions, please clarify. Overall, I think if there were 4 configurations in total it would be more systematic where only one thing would change at a time, 1: inventory-infinitely wide-default 2: inventory-finite radius-default 3:tls-finite radius-default 4: tls-finite radius-tls

**This intermediate setup will be included in the global sensitivity analysis and should not have been omitted in the first place.**

L237-242: As much as I liked the process-based perspective, a sensitivity analysis (running the model with varying parameter combinations drawn from their priors to see how much change they cause on model outputs) would also be warranted here to formally show these parameters are indeed constrainable by the fluxes. Also the authors might be missing some other important model parameters (although there may be many parameters that can be calibrated as authors suggest in the discussion, models are typically most sensitive to maybe a dozen or so). I.e. calibration might be pushing SLA and Vcmax to different values in the

parameter space under different configurations, but in fact if other parameters were included in the calibration it may have not been the case. Besides, other aspects of a proper calibration protocol are skipped here. For example, after determining to target these two parameters, authors could vary these parameters in their prior ranges and plot a likelihood surface (if they had done a global sensitivity analysis this would have come for free). This would have revealed the trade-off (negative correlation) before the calibration and would further implicate the need for either more informative priors (see below), or even not targeting one of these parameters in the calibration. I would have understood if authors, so to speak, enforce equifinality and use TLS to resolve it, but that has not been the case in the end (authors only report differences, don't really conclude -validate- which was more accurate). Instead, authors exacerbate the equifinality issue by choosing correlated parameters and uninformative priors only to confirm low identifiability (L390) and mention TLS' potential to discriminate without actually doing so. To sum up, I have three suggestions for the authors: 1) perform a global sensitivity analysis to at least identify other important parameters, even if they decide not to calibrate them it could help discussion, 2) try to repeat the analysis with more informative priors, 3) elaborate on their calibration results (some suggestions below) and strengthen their conclusions (be less vague).

**We globally agree with this comment and the proposed new global analysis should address most of the reviewer's issues. Note that we will now include more parameters than just SLA and V$_{cmax}$ based on previous ED2 studies and will use more informative priors based on available trait data. Furthermore, the model performance with and without TLS data will be evaluated with the independent datasets mentioned above.**

L246: GPP is not measured but a derived (modeled) quantity, at least as opposed to other carbon (net ecosystem exchange) and water (latent heat) fluxes. How the uncertainties were affected in this case, how was that accounted for in the calibration?

**We will add a discussion about the uncertainties associated with the way GPP is modelled from the raw eddy-covariance data. In addition, we will provide more information about the origin and the treatment of the original data in the material and methods, as suggested by the first reviewer.**

L254: Sampled how? From marginal or joint posterior distributions? Please clarify.

**From the joint posterior distribution but this is no longer relevant in the new analysis that we propose.**

Table 3 and Figure S3 Vcmax units are different from L144 and Fig 5, please reconcile.

**We will make sure all units are consistent in the new version ($V_{cmax}$ has $\mu mol_c\ m^{-2}\ s^{-1}$ units)**

L255 and Table 3: Why were the priors chosen to be uniform? Are values like 5 really equally likely as 30-40 or is 60.5 impossible for Vcmax? I believe given many observations and prior knowledge about these parameters more informative priors could have been chosen, which in return could have reduced the equifinality problem. Please consider distributions other than uniform.

**These prior distributions will now be constrained by trait data, whenever those data exist.**

Figure 3: Looking at the figure, hard to tell without playing with the raw data, but it almost looks like there could be two lines fitted to Acer (late hardwood) and others (mid hardwood) reinforcing the point about exploring two PFTs above.

**We will test that and classify the tree species into the PFTs accordingly (and also based on trait data and previous classifications published in the literature such as the one provided by the reviewer).**

L263: If the fitted parameter values of the lines on Figure 3 are on Table 2, please already say so in this paragraph. Also Fig3 caption can refer to Table 2.

**We will make these links in the new version.**

L267: Although, maybe it is worth noting that both are performing badly at the tails. Please consider providing the residual plots for Figure 3 fits in the supplement.

**Good suggestion, we will provide the residual plots for the allometries in the future version of the supplement (with one and several PFTs).**

L270: I don't mind these figures being in the supplement, however, I felt like this should have been the first thing reported in the results. How well TLS do with respect to inventory, before moving on to allometries.

**Agreed. And based on reviewer #1' suggestion this will probably move to the main text.**

Figure 4: It is surprising how big of a difference infinite versus finite crown configurations makes. Was this documented before in previous ED2.2 studies? Is it appropriate to spinup both configurations for 100 years? Can it be that the FC configuration needs to be run longer? Also I believe each of these bars represents a single realization from the model, is that right? I would be curious to see if slightly different NBG initializations in an ensemble mode could have provided a different picture (i.e. Fig 4 but with error bars where some configurations may manage to get into the right ballpark). Plus, NBG ensembles (with different initial conditions and Vcmax/SLA parameter combinations) could provide an additional quantification of the uncertainty reduction (their ensemble widths can be compared to their IC counterparts which were constrained by TLS). Besides, how would the tree size distribution look like if the authors have used more than just one PFT in these simulations (further point, as noted by the other reviewer, it was not clear if the seedlings were MH-only)?

**All these issues will be solved with the new analysis (including one *vs* multiple PFTs, ensembles from near bare-ground conditions, and the quantification of the uncertainty reduction). With the new ensemble runs, we will be able to assess if slightly different NBG initializations could have provided different pictures of the tree size distribution. The large impact of finite *vs* infinite crown representation was documented only recently in a publication (Shiklomanov et al., 2019). It is a well-known 'issue' in the ED2 community but probably not outside (yet see Fisher et al., GCB, 2018).**

L281-282: This is a clear result, however, (looking at Table 4, as mentioned above) I would be curious to see a NBG-infinitely wide-TLS configuration with TLS informed allometric coefficients. Does the infinitely wide configuration not use the same allometric equations? These models can be highly non-linear, and the response of NBG-FC with and without TLS allometries could be different than the sensitivity of NBG-infinitely wide with and without TLS allometries. And if it is the same result, it would only strengthen this finding.

**The new model analysis will account for that missing configuration.**

L293, Figure 5:  After referring to Fig 3, please try to make it clear here that you are back to referring to Fig 5 here. E.g. "The large variability around those mean relative changes (Fig 5, error bars) ..." Also state in Fig 5 caption what do error bars represent.

**Figure 5 will no longer exist in the revision.**

L297: Yet, I'd somewhat expect larger aboveground woody biomass could also result in bigger trees -> less understory PAR. Does it imply problems in model structure?

**It definitely relates to problems in the model structure (the woody tissues do not account for light interception). It will be discussed in the revision.**

L329: Unfortunately, there is no Table 5.

**We are sorry that this table somehow disappeared during the submission process. It will be included in the next revision.**

L331-334: IC-TLS uses both the DBH distribution and the allometric coefficients informed by TLS. So I assume it was able to capture Figure3 - leaf biomass relationship very well? Sounds like it also produced LAI values in the right ballpark. The link between leaf biomass and leaf area is through the SLA in the model (L238: SLA is used to convert the leaf biomass into leaf area), and SLA posterior of IC-TLS is agreeing with the CWM. So it almost looks like this configuration gives the right answers for the right reasons for these variables and parameters. And if IC-FC is producing the same leaf area as IC-TLS but have different SLA, it must be missing leaf biomass? I'm trying to see if authors could discriminate a bit more explicitly about performances of different configurations here, please consider elaborating as such.

**In the new version of the manuscript, we will make sure which configurations perform better for every single independent observation (eddy covariance fluxes, leaf area, basal area, growth etc.). Doing so, it will be clear how TLS data improves model performance and reliability.**

L343: Does this contradict or how is it related with the finding mentioned before on L281-282: when NBG is concerned model structure had a bigger impact than allometries on tree size distribution? Also although I couldn't see Table 5, I have a feeling that it will be hard to digest. I recommend authors consider a figure instead or in addition.

**This sentence (L343) does not contradict previous findings because in that case we were investigating the allometric parameters only and we did not compare the relative process and parametric errors. This is something we will investigate with the new analysis. The**

**figures and tables are very likely to change significantly but we will make sure they are digestible or we will increase the number of graphic elements accordingly.**

L381-382: Could you be more specific here? The statement is too vague. NBG with TLS informed allometry didn't do any better for capturing the tree size distribution (they also did bad for the ecosystem variables L346). So informing allometry alone was not enough. Is TLS more useful when it is used for prescribing the initial conditions or what? How does this agree with studies in the literature? E.g. does this mean initial condition uncertainty is a bigger problem than allometric uncertainty?

**In the new version, we will be able to demonstrate the benefits of TLS to more adequately represent the modelled ecosystem. As explained above, to do so we will compare the performance of the different tested configurations against independent validation datasets. By partitioning the overall uncertainty, we will also quantify the relative benefits of prescribing the initial conditions, the model structure or the allometric equations.**

L385-386: I don't know if it is striking, but it was expected given the trade-off and uninformative priors.

**Fair enough. This problem will be solved by informing the priors with the trait data in the new analysis.**

L388-389: "Very different" but how? Again very vague. Was a particular one any better?

**This will clearly emerge from the new analysis.**

L395-396: But did it in this study? Does this mean the authors trust IC-TLS posteriors on Fig 7 more? Also please see my comment above for lines 331-334, and try to be more specific. If you did discriminate between equifinal model versions, say it here which did better.

**We will in the new version (see our comments above).**

L407: But aren't there more formal ways to deal with this? E.g. one could start the model from the past (when flux data is available) with more uncertain IC even if they don't know about the forest structure and composition a decade ago, and then calibrate the model with past data, continue simulations in time and assimilate more recent inventory data to constrain the states? In fact, the Thomas et al 2011 paper cited by the authors already have some useful values for conditions 10 years ago. Furthermore, the Butt et al. citation implies there was a tree census in 2008? (I merely clicked the link)

**The recurring forest inventories will be added to the list of validation datasets used to discriminate the model performance.**

L412: While it is true that it would increase the overall complexity of the study, I'm not sure it sufficiently justifies simulating one PFT when at least Acer and Quercus are concerned. Could more informative priors be chosen when more distinct PFTs are used? There were also numerous occasions mentioned above where using multiple PFTs could potentially remedy some of the shortcomings. At least without demonstrating it, I'm afraid this argument remains unconvincing.

**This will be solved by (i) testing more than one PFTs and (ii) using more informative priors for some of the traits**

L418: This statement, although true, seems rather irrelevant for the conclusion of the present study as both SLA and Vcmax are measurable. In general, until the last three sentences, the conclusion reads like an introduction and needs to be tailored towards the study more. I'd recommend starting from what you demonstrated, then telling what the implications of your findings are, how well your results aligned with your prior expectations, if your methodology was adequate, if you got new insights / new ideas for future steps and so on.

**We will make sure the conclusion is tailored towards our findings. As we expect those to be significantly impacted by the new analysis, we prefer not to give too much detail here about its future content.**

L422-425: Apologies for repeating myself but I overall think the reporting was rather inconclusive as to whether the TLS informed model was indeed more reliable or able to discriminate between equifinal model versions. In other words, yes, TLS-informed results were different but were they more realistic? What was the independent validation? Which configuration got the right answers for the right reasons? Reader has to work really hard to figure it out. You could further provide your concluding recommendation regarding how TLS is best utilized.

**All those fair points should be hopefully addressed by the new conclusions emerging from the global analysis.**

---

## Author Response (AR1)

**Response to anonymous referee # 1**

The concept for this study is strong, using TLS to constrain forest structure and function in the ED2.2 model, follows a decade and a half's work on using remote sensing to constrain predictions made by ecosystem models (in reducing process and initialization errors). While this idea is worth publishing, the execution is not clear, the structure of the study needs improvement, and the actual constraining of the ED2.2 model is adequately done. Concerning this last point. Essentially you want to know how well your TLS-constrained ED2.2 simulations has fared compared to Ground-based-initialized ED2.2 simulations and compared to bare-ground simulations. To assess the improvement you need to compare all 3 of these simulations to observed data (like GPP, plot basal area changes, and/or growth and mortality dynamics). You need to do this for both TLS-structure and TLS allometric improvements.

**We first would like to thank Reviewer #1 for their positive review of our work. We were pleased to see that they share our opinion that TLS has strong potential to successfully constrain forest models and that such an idea is worth publishing In Geoscientific Model Development. After reading both reviews, we decided to thoroughly reshape our study in order to improve the structure of the manuscript and enhance the execution of the research idea. The concepts, data, and methods remain essentially the same but were reshaped and presented in a different manner.**

**As highlighted by both reviewers, what we are interested in is (i) the sensitivity of the ED2 model to parameters and processes that can be constrained/defined via TLS-derived observations and (ii) how model simulations constrained with TLS data compare to ground-based initialised simulations, bare-ground model runs and independent observations. In order to investigate this and address the main comments of both**

reviewers, we replaced the current set of analyses by an overall global sensitivity and variance decomposition analysis applied to different model configurations (near-bare ground, inventory-initialised and TLS-informed). This new analysis encompasses the three included in the previous version of the manuscript and extends them. For each model configuration, we ran ensemble simulations (N = 500) and decomposed the total variance into its components (model structure and parameters). This allowed us to answer three research questions that were already present in the previous version of the manuscript but were not explicit, and so only partly identified and answered. Those research questions are:

(i) What is the total model uncertainty of the ED2 model for the specific site that we try to mimic in silico, and what are the contributions of the different sources of uncertainty?

(ii) Is the total model uncertainty reduced when constraining the model with TLS observations and dothe primary sources of uncertainty remain the same?

(iii) Does the use of TLS data improve the model performance?

Model ensembles were started from TLS-derived inventories (TLS configuration), from near bareground conditions (NBG) or field inventories (Census). The importance of the model structure was tested in the different configurations by changing the crown representation (finite or infinite), the number of simulated PFTs (see below for details), the type of radiative model used and the trait plasticity (the summary of configurations is listed in Table 2 of the manuscript) . The impact of model parameters was quantified by changing a larger set of parameters than just SLA and $V_{c,max}$ as suggested by the second reviewer. We included the parameters that were shown to drive a significant fraction of the model uncertainty in previous studies (see e.g. Dietze et al., 2014 and Shiklomanov et

al., 2019) and not only the ones for which we have trait data available. Hence, we could separate the parameter uncertainty into the contribution of TRY-constainable parameters, allometric parameters and other ED2.2 parameters. In the TLS configuration, the allometric parameters were constrained to TLS data while the prior distributions of those parameters remained uninformed in the NBG and Census configurations.

Those ensembles served to quantify the total model uncertainty and partition it into its components. Doing so, we also could estimate the benefits of TLS to reduce the total model uncertainty and how its drivers change when some processes and parameters were constrained by TLS data. Finally, model outputs were compared to the datasets previously used to calibrate some of the model configurations (mainly eddy covariance fluxes, and LAI). Doing so, we were now directly able to compare the performance of the different configurations.

There are many more comments in the text attachment below.

We thank reviewer #1 for this detailed review, which is very useful to correct all minor points. As the manuscript changed significantly, we do not provide here a detailed response for every single comment. Instead, below we repeated and answered only the comments that remained relevant for the new version of the manuscript and the new analyses.

Detailed comments and response

L25: "... at a temperate forest site" As it is a single site, isn't it meaningful to say what it is?

**We now name the site (Wytham Woods, UK) starting from the abstract**

L27: "... productivity ..." gross primary productivity? net primary productivity? leaf area production?

**The global sensitivity and variance decomposition analyses were performed on several model outputs, including gross primary productivity.**

L29: "the imposed openness" do you mean the gap fraction here?

**By "imposed openness" we meant "the representation of the canopy (infinite or finite crown areas)". We do not use this term anymore in the new version of the manuscript.**

L35-39: "We conclude … and reduce their overall uncertainties" Have you validated you new TLS-configured ED2.2 runs? Do you have tested reductions in uncertainty?

**Validating the TLS-informed runs with independent datasets (including eddy-covariance data, basal area changes, etc.) was one of the objectives of the new analysis and is now presented in the new version of the manuscript. We showed that the model uncertainty of the model outputs is reduced when simulations are constrained with TLS data, and even more when both TLS and TRY data are used. The TLS also exhibits good performance for reproducing the seasonal cycle of GPP and predicting the ecosystem LAI.**

L165-166: "All trees inventoried in Wytham Woods were classified as mid-successional temperate deciduous trees" Are you sure. Sycamore can be considered a shade-tolerant tree (late successional), maybe hazel. You need to go through each and every species to see which are early, mid and late successional species. Identifying them all as mid-successional needs a good explanation.

**As the manuscript was initially seen as a simple sensitivity analysis of the ED2 model to the parameters and processes that could be constrained to TLS data, we wanted to simplify the analysis as much as possible and hence decided to simulate a single PFT. Incorporating more than one PFT would have made the whole analysis way more complicated to present, as well as for rendering the outputs to the readers. Yet, with the new global analysis, it was possible to take this complexity into account by simulating one or several PFTs and include that uncertainty into the model structure error. The tree species mapping to the model PFT was achieved using previous classifications of the literature and the allometric relationships derived from TLS. Our analyses showed that due to the compensatory effect, the contribution of the number of simulated PFTs to the overall model uncertainty was only a small fraction of the total variance (see Figure 4 in the new version of the manuscript) .**

L191: "i) near-bare ground initial conditions (i.e. seedlings only" Is this with Mid-Successional Hardwoods seedlings only?

**In the previous runs, it was indeed with Mid-Successional Hardwoods only. In the new runs, it is with Mid-Successional Hardwood trees or a combination of Mid- and Late-Successional Hardwood trees (see comment above).**

L193-194: "that were allowed to fuse along the simulation" Unclear. Delete?

**This part of the sentence was deleted.**

L194-195: "Simulations were run … dataset (Viovy 2018)" We need more information on the initializations. How long was the bare-ground simulation run for? How did you cycle the years of the met data? Which PFTs did you use? What soil data were used at the site? What is the resolution of your met data? What are the structural attributes of the ground vs TLS initial forests...i.e. Basal Area, DBH, LAI, TLS occlusion etc.

**Those pieces of information were provided in the new version of the manuscript. The NBG simulations were run for 100 years (the approximate age since last large-scale disturbance) and forced with the corresponding years of the CRU-NCEP dataset. For the soil data, we used soil texture analyses from the literature. We also now compare the tree size distribution of the inventory vs TLS (Supplementary Figures S4 and S5).**

L214-216: Can you have results for this analysis and comparisons to ground-inventories?

**In the revised version, all near bare-ground runs are compared to ground inventories as an independent validation (Figure 6).**

L231: "Parameter data assimilation and model equifinality (Analysis III)" Rather than equifinality, this method appears to be about parameter optimization. Perhaps I am wrong.

**We agree with the reviewer's comment. Analysis III is no longer included in the new version of the manuscript (no more parameter optimization was done in the new analysis).**

L261-264: I can't find these numbers in the figures

**Those numbers (re-computed as RMSD) were added in the new version of Figure 3 (now Figure 2 and supplementary Figure S2)**

L265-271: These results are not clear. It is assumed you are describing Figure 3, but numbers comparing TLS and ED2 are not clear.

**We now explicitly refer to the appropriate figure (now Figure 2).**

L273-275: TLS and ground differences are important here and should not be in the supplement. Valuing the use of TLS in ED2.2 needs a rigorous comparison to ground-inventory data and initializations.

**We expanded the discussion about the differences between TLS and ground-inventory and improved the figures comparing the TLS and the inventory tree size distribution**

**(Supplementary Figures S4-S5). Yet we decided to keep those in the supplementary file as it is not the main focus of this paper, and what this analysis reveals is essentially an almost perfect correlation between TLS and ground-inventory data.**

L287: Start off making reference to the Figure you are about to explain

**We made sure we properly start off making references to the appropriate figures in the new version of the manuscript.**

L304: It is still unclear why you are doing this step, and why you are not including the new optimized results in the sensitivity analysis (Figure 5). Also it seems counter-intuitive that IC-Default Vcmax value of 47.3 compared to 17.5 would not largely affect GPP!

**The $V_{cmax}$ definitely influenced the modelled GPP but the effect of the parameter was compensated by the reduction in SLA resulting from the optimization, which eventually led to similar GPP. In the new analyses we replaced the optimizations by a global sensitivity analysis and compared the (sub-)ensembles with the (now) independent eddy-covariance data.**

Figure 7: Confusing figure. What this tells me is that the TLS and FC initial conditions are not that close, especially with the results in SLA. How useful is this optimization, and why do this if you are not going to use it in the sensitivity (Figure 5), or in ultimately comparing the improvements of TLS vs ground-inventory initializations against carbon fluxes?

**The optimization analysis was removed from the new version of the manuscript.**

Figure 7 (bis) What does this mean?/What is this dot?

**The dots were the data from the TRY database. Yet, this figure was removed from the new version.**

L355: There is no Table 5

**We are sorry that Table 5 was missing in the manuscript. It somehow disappeared in the submission process. Those data are now available in the new version (Table 6).**

L359-365: It may be expected that better site level descriptions of BLeaf allometry will affect the LAI and GPP, and that better descriptions of Bdead will affect Woody biomass. Rather than sensitivity, what about overall improvement to LAI, GPP, Biomass??

**Following yours and reviewer #2' suggestions, we now included this analysis in our study. Site-level leaf and woody biomass allometries improved the performance of the model for the simulated LAI, GPP, and AGB.**

L383-384: Not sure you have done showing the effective improvement in model performance.

**The new set of analyses allowed us to conclude about the effectiveness of using TLS data to constrain the model uncertainty and improve model performance. In the new version we show a strong reduction in uncertainty and a significant improvement of model performance thanks to the TLS data for multiple state variables independently observed (LAI, GPP).**

L397-398: From your results, this 'equifinality' issue could be demonstrated without TLS, just by optimizing using bareground and ground-inventory initializations. Furthermore, the usefulness or added value of this equifinality/optimization excercise is still not clear.

**The optimization exercise is no longer present in the new version of the manuscript and equifinality still emerges from the large ensemble runs.**

L403-L404: You could use recurring forest inventories collected at your site as an additional calibration dataset

We did not use successive inventories from Wytham Woods in the first version of the manuscript. Since we could not get access to such information and we could not find any relevant data in the literature, our analysis only relies on a single forest inventory.

**Response to anonymous referee # 2**

The manuscript by Meunier et al., uses TLS data to inform coefficients of an ecosystem model's allometric equations and initial conditions, quantifies its impact, as well as testing influence of TLS information on model calibration. While the study is well thought out and generally well-written and visualized, there are some issues with both modelling and calibration protocols (in terms of both technicality and clarity). Also the manuscript remains somewhat inconclusive about the superiority of TLS-informed model predictions, or at least if that wasn't the case the manuscript needs to be revised to clearly present it as such. It's a pity Table 5 was not available for the review process. Overall, I think the study would be of interest for the community and worth publishing, however, I would strongly recommend tackling the technical issues raised by both reviewers. Line numbers below refer to the author's preprint.

**First, we would like to thank reviewer #2 for their thorough assessment of our manuscript. We believe that the comments raised were fair and contributed to improving the overall quality of the study. The suggestions fall in line with the ones of the first reviewer and we agree the results were not presented in the clearest way.**

**As explained in the response to reviewer #1, we decided to reshape the manuscript around one central analysis, which replaced and extended the previous three. The concepts, data, and model remained essentially the same but were analysed and presented in a different manner. The new analysis assesses the global sensitivity of the model and evaluates its performance when the simulations are constrained or not by TLS data. More precisely, we ran large ensemble simulations under different model configurations (near bare-ground, inventory-initialised and TLS-informed), and partitioned**

the overall uncertainty into its main components (model structure, and model parameters) for each configuration.

Model runs were initialised from near bare-ground (NBG), field inventory (Census) or TLS-derived size distribution (TLS). For the model structure, we tested the impact of the crown representation (finite or infinite), the choice of radiative transfer model, simulating more than a single PFT, and the trait plasticity (see Table 2 in the manuscript). Additionally, we increased the number of parameters to be tested to include those that were shown to be significantly contributing to the overall model uncertainty in previous ED2 studies (see Table 4, 13 parameters in total). These global sensitivity and variance decomposition analyses allowed us to investigate the following research questions that were somehow present in the previous version of the manuscript but not clearly identified nor fully answered:

(i) What is the total model uncertainty? What are the contributions of the different sources of uncertainty? And how do they change along the simulation?

(ii) Is the total model uncertainty reduced when constraining the model to TLS observations? Do the primary sources of uncertainty remain the same?

(iii) Does the use of TLS data improve model performance?

To answer the last question, we used independent datasets that were previously used for model calibration (mainly eddy-covariance fluxes and LAI) as validation datasets and compared how the ensembles reproduce those. Most of those data were included in Table 5 (now Table 6) and we are sorry that it disappeared during the submission process. We are very confident that such a reshape of both the study and the manuscript significantly improved the quality and clarity of the protocol and now clearly shows the positive impact of the use of TLS data on model uncertainty and performance.

**We also thank reviewer #2 for the detailed review, which was very useful to correct all minor points. However, as the manuscript changed significantly, some of the comments were no longer relevant. For that reason, below we only answered the comments that were still relevant for the new version of the manuscript and the new analysis.**

Title: As mentioned in the general comment above, the manuscript is rather inconclusive about the reliability of TLS informed model predictions. Even the abstract reports only the sensitivity of the results to model configuration and TLS information. Hence, it feels as if the title would reflect the study more closely if it was revised to something along the lines of "Sensitivity of ED2.2 forest ecosystem simulations to TLS informed/constrained structure and functions" (as also presented by the authors on L95 and L195).

**We agree that we mainly tested the sensitivity of the model simulations to TLS-informed structure and functions in the previous version of the manuscript. Yet, we think that the new version of the model analysis allowed us to be more conclusive about the reliability of the TLS-informed simulations, and the increased performance when integrating TLS data. Therefore we would prefer to keep this manuscript title.**

L28: "imposed openness" do you mean the FC configuration here? If yes, please revise to explicitly say "model configuration that imposes finite canopy radius dramatically influenced..."

**We indeed meant the crown representation here (finite or infinite) but we no longer use this term in the new version of the manuscript.**

L33-34: After reading the manuscript, I wasn't quite left with a conclusion about the most adequate model structure. If you identified it, why not say it in the abstract explicitly.

**We agree that the previous set of analyses did not allow to identify the most adequate model structure because we lacked independent datasets (eddy covariance fluxes were**

**used to calibrate the model, and LAI observations were derived from the same TLS datasets which served to parameterize the model). Reshaping the analysis as described above allowed us to reach such a conclusion (TLS configuration with TRY constraints) and we now repeatedly state it in the paper, e.g. in the abstract.**

L81: Somewhere around this paragraph I would have expected a brief introduction about other (e.g. airborne) lidar studies with TBMs as well. Especially given that studies exist directly with the ED model, Hurtt et al. 2004 (https://doi.org/10.1890/02-5317), 2019 (https://doi.org/10.1088/1748-9326/ab0bbe), Thomas et al., 2008 (https://doi.org/10.5589/m08-036). I think this could benefit the discussion as well, e.g. what did the authors build upon the previous lidar-ED2 studies? or they can draw parallels to this study.

**We thank the reviewer for pointing out these very relevant references that were unknown to us. We now included these references in the introduction and discussion of the new version.**

L136-140: Appreciated the length authors went with extracting the data. However, this paragraph would benefit from further information on the overall quality of the data: what the frequency of the data is (daily, sub-daily?), how it was filtered, QA/QC'd, how the GPP was derived, what the accuracy of data retrieval from Plot digitizer software is, if there are known issues with the time series that could affect the calibration and so on.

**Unfortunately, there is not much that we can add on top of what was already described in the original publication reporting those data. Therefore, we now explicitly refer the reader to the Thomas et al. (2011) paper for more information.**

L152: I'd like to point out that the authors themselves avoid using the word "validation" here, which again reinforces my comment about inconclusiveness. In case you decide to strengthen the paper's conclusions, at least consider the word "assessment" here.

**The trait data now serve to inform the parameter distribution before the global sensitivity analysis. Model performance was now assessed with other independent datasets (including the eddy covariance fluxes) that were previously used to calibrate some of the model configurations. In that sense, we now use those observations as validation datasets.**

L164: Agreed with the other reviewer. Why was all classified as mid-successional pft in ED? I agree that each species, at least the five on Figure 1, needs reasoning as to which PFT they were mapped to and why. Please also provide citations for mappings when possible e.g. see supplementary on https://onlinelibrary.wiley.com/doi/full/10.1111/j.1365-2486.2011.02477.x where Acer is LH, Quercus is NMH. Admittedly, using multiple PFTs would complicate the reporting as authors are currently only concerned with a single set of allometric parameters, but worth exploring. Also, even if the authors decide to continue with a single PFT after revisions, they should emphasize already here that this is an over-simplification which could help prevent misuse by others referring to this study in the future.

**The first version of the manuscript mainly aimed at assessing the sensitivity of the ED2 model to parameters and processes that could be constrained by TLS data. At this stage we wanted to simplify the model complexity as much as possible to render a clear and simple message. Increasing the number of PFTs, while important to simulate the complexity and diversity of acquisition strategies of the tree species, would have in our opinion made our analyses and conclusions fuzzier and more difficult to grasp. In the new manuscript, we**

**overcame the issue by incorporating the number of modelled PFTs in the model structure uncertainty and doing so we now quantified the uncertainty associated with the simplification of simulating a single PFT in the new analysis. The tree species mapping to the model PFT was achieved using the allometric relationships derived from TLS, as well as previous classifications from the literature, as suggested by the reviewer. The best mapping was indeed to classify Acer as Late-successional Hardwood trees and all other species as Mid-successional Hardwood trees.**

L192: Agreed with the other reviewer. Please provide more details or point to the initialization/settings files of ED2.2 specifically if you have deposited them to the repository cited at the end (you could have a supplementary table telling which initialization/settings files went with which experiment or populate the readme file on the repository) .

**We now included the settings files of the model in the Zenodo repository.**

Figure 2 is great, but I'd call Analysis III: Bayesian calibration instead of data assimilation to be more precise, or at least continue using "parameter data assimilation". Also for analysis I, did you use TLS to inform structure directly? Looking at Table 4 it's only allometries. Allometries in return affect the structure but if I saw only allometries in that box, it would have helped me follow the study better.

**We agree with the reviewer that "parameter data assimilation" or "Bayesian calibration" was more appropriate. This analysis no longer appears in the revised version of the manuscript and Figure 2 was removed.**

L202-203: What do you mean by "to assess the relative importance of TLS we compared it to field observations"? Does this exercise result in Fig S1 and S2? Isn't it then better to call this ground-truthing or validation of TLS? Please clarify.

**Indeed this corresponds to Figure S1 and S2. We rephrased this sentence:**

**"To assess the relative importance of TLS for the model initialisation, we compared the tree size distributions obtained from the field inventory and from the TLS data and computed the absolute and relative differences between both DBH distributions (ground-truthing of TLS)."**

L207: Could you already explain here if 100 years spinup is enough, especially considering that the actual age of the forest is much older? I know of other models running much longer spinups (e.g. 500 years), please motivate the reader if 100 years is appropriate.

**The 100 year-long run does not correspond to a true model spinup but rather to the approximate age of the forest (it is the last time since large-scale disturbance occurred to Wytham Woods). So if the model was perfect, the size distribution as simulated after 100 years should resemble the actual observation. However, 100 years are sufficient for the model to grow virtual trees larger than the largest trees observed in Wytham Woods.**

L214: Looking at Table 4, how about NBG-infinitely wide-TLS setup? See comment below regarding having another control for impact of TLS informed allometries.

**We did not include the NBG-infinitely wide-TLS setup because TLS data allowed us to constrain the crown area allometry.**

L221: Why not explicitly state in what order these changes and combinations were introduced as this might also help following the incremental effect discussion. Listing configurations for 16 runs is not that much, could be also in the supplementary.

L226-228: Agreed with the other reviewer on quantification of indirect effects. I think listing all the configurations for the mentioned 16 runs will help. I assume authors performed a factorial design here but it is not clear which combinations went with which.

**For both previous comments: the new global sensitivity analysis does not use this factorial design anymore so we got rid of those issues while keeping the main results of the**

**sensitivity analysis (relative contribution of the allometries to the overall model uncertainty).**

L231: "parameter optimization by Bayesian data assimilation" -> Authors could consider using "Bayesian parameter data assimilation" here as well to be more clear. Or better yet, "Bayesian calibration of model parameters".

**We agree with the reviewer's comment. Since the analysis was reshaped, this section title was removed from the main text.**

L235: Looking at Table 4, it feels like there needs to be another intermediate setup: inventory-finite radius-default, is there a particular reason why authors omitted this configuration? Also this sentence on L231-L233 suggests this configuration was included but Table 4 does not mention this configuration: "The model configurations included a default model version (default allometric parameters, infinite crown area), and a finite crown representation (default allometric parameters, finite crown radius), *which were both initialized with field inventory data*" I believe, according to this sentence, Table 4 second to last column should read "inventory" for initial conditions, please clarify. Overall, I think if there were 4 configurations in total it would be more systematic where only one thing would change at a time, 1: inventory-infinitely wide-default 2: inventory-finite radius-default 3:tls-finite radius-default 4: tls-finite radius-tls

**Following the reviewer's comment, the new analysis redefined those configurations.**

L237-242: As much as I liked the process-based perspective, a sensitivity analysis (running the model with varying parameter combinations drawn from their priors to see how much change they cause on model outputs) would also be warranted here to formally show these parameters are indeed constrainable by the fluxes. Also the authors might be missing some other important model parameters (although there may be many parameters that can be

calibrated as authors suggest in the discussion, models are typically most sensitive to maybe a dozen or so). I.e. calibration might be pushing SLA and Vcmax to different values in the parameter space under different configurations, but in fact if other parameters were included in the calibration it may have not been the case. Besides, other aspects of a proper calibration protocol are skipped here. For example, after determining to target these two parameters, authors could vary these parameters in their prior ranges and plot a likelihood surface (if they had done a global sensitivity analysis this would have come for free). This would have revealed the trade-off (negative correlation) before the calibration and would further implicate the need for either more informative priors (see below), or even not targeting one of these parameters in the calibration. I would have understood if authors, so to speak, enforce equifinality and use TLS to resolve it, but that has not been the case in the end (authors only report differences, don't really conclude -validate- which was more accurate). Instead, authors exacerbate the equifinality issue by choosing correlated parameters and uninformative priors only to confirm low identifiability (L390) and mention TLS' potential to discriminate without actually doing so. To sum up, I have three suggestions for the authors: 1) perform a global sensitivity analysis to at least identify other important parameters, even if they decide not to calibrate them it could help discussion, 2) try to repeat the analysis with more informative priors, 3) elaborate on their calibration results (some suggestions below) and strengthen their conclusions (be less vague).

**We generally agree with this comment and the proposed new global analysis addressed most of the reviewer's issues. Note that we now included more parameters than just SLA and V$_{c,max}$ for the parameter uncertainty based on previous ED2 studies and used previously defined priors and posteriors based on available trait data (see Table 14 with 13**

**parameters). Furthermore, the model performance with and without TLS data was evaluated with the independent datasets mentioned above.**

L246: GPP is not measured but a derived (modeled) quantity, at least as opposed to other carbon (net ecosystem exchange) and water (latent heat) fluxes. How the uncertainties were affected in this case, how was that accounted for in the calibration?

**We added a discussion in the study limitations section about the uncertainties associated with the way GPP is modelled from the raw eddy-covariance data:**

**"In addition, we know that GPP is not directly observed but rather a derived (modelled) quantity, at least as opposed to the net ecosystem exchange of carbon and the latent heat flux of water. Unfortunately, we could not access water flux raw data nor were they reported in publications that we knew of. GPP uncertainties were also not quantified in the original publication of Thomas et al. (2011). While NEP values were reported, validating the model simulations with those values would have biassed our analyses as we could not constrain respiration parameters with data."**

L254: Sampled how? From marginal or joint posterior distributions? Please clarify.

**They were sampled from the joint posterior distribution but this is no longer relevant in the new analysis that we ran.**

Table 3 and Figure S3 Vcmax units are different from L144 and Fig 5, please reconcile.

**We made sure all units are consistent in the new version ($V_{c,max}$ has $\mu mol_c$ $m^{-2}$ $s^{-1}$ units)**

L255 and Table 3: Why were the priors chosen to be uniform? Are values like 5 really equally likely as 30-40 or is 60.5 impossible for Vcmax? I believe given many observations and prior knowledge about these parameters more informative priors could have been chosen, which in return could have reduced the equifinality problem. Please consider distributions other than uniform.

**These prior distributions were redefined and/or constrained by data, whenever those data exist (SLA and $V_{c,max}$).**

Figure 3: Looking at the figure, hard to tell without playing with the raw data, but it almost looks like there could be two lines fitted to Acer (late hardwood) and others (mid hardwood) reinforcing the point about exploring two PFTs above.

**We tested the possible classifications and retained the one that minimised the Watanabe information criterion. It indeed led to classifying Acer as late hardwood and all other species as mid hardwood. In any case, the number of considered PFTs had a very limited impact on the model uncertainty.**

L263: If the fitted parameter values of the lines on Figure 3 are on Table 2, please already say so in this paragraph. Also Fig3 caption can refer to Table 2.

**We made these links in the new version.**

L267: Although, maybe it is worth noting that both are performing badly at the tails. Please consider providing the residual plots for Figure 3 fits in the supplement.

**We thank the reviewer for the suggestion. We now provide the residual plots for the allometries in the new version of the supplement (with one and several PFTs), see Supplementary Figure S3.**

L270: I don't mind these figures being in the supplement, however, I felt like this should have been the first thing reported in the results. How well TLS do with respect to inventory, before moving on to allometries.

**Those results were moved to be the first thing reported in the results. However, we kept the figures in the supplement because as they essentially show a perfect correlation between TLS and the ground-inventory data, the figure is not especially information-rich.**

Figure 4: It is surprising how big of a difference infinite versus finite crown configurations makes. Was this documented before in previous ED2.2 studies? Is it appropriate to spinup both configurations for 100 years? Can it be that the FC configuration needs to be run longer? Also I believe each of these bars represents a single realization from the model, is that right? I would be curious to see if slightly different NBG initializations in an ensemble mode could have provided a different picture (i.e. Fig 4 but with error bars where some configurations may manage to get into the right ballpark). Plus, NBG ensembles (with different initial conditions and Vcmax/SLA parameter combinations) could provide an additional quantification of the uncertainty reduction (their ensemble widths can be compared to their IC counterparts which were constrained by TLS). Besides, how would the tree size distribution look like if the authors have used more than just one PFT in these simulations (further point, as noted by the other reviewer, it was not clear if the seedlings were MH-only)?

**All these issues were solved with the new analysis (including one *vs* multiple PFTs, ensembles from near bare-ground conditions, and the quantification of the uncertainty reduction). With the new ensemble runs, we were able to show that slightly different NBG initializations could provide different resulting tree size distributions (new Figure 6, keeping in mind that that is drawn from the "vegetated" simulations only and many others showed zero vegetation after 100 years). The large impact of finite *vs* infinite crown representation was documented only recently in a publication (Shiklomanov et al., 2019). It is a well-known 'issue' in the ED2 community but probably not outside (yet see Fisher et al. GCB, 2018).**

L281-282: This is a clear result, however, (looking at Table 4, as mentioned above) I would be curious to see a NBG-infinitely wide-TLS configuration with TLS informed allometric

coefficients. Does the infinitely wide configuration not use the same allometric equations? These models can be highly non-linear, and the response of NBG-FC with and without TLS allometries could be different than the sensitivity of NBG-infinitely wide with and without TLS allometries. And if it is the same result, it would only strengthen this finding.

**The new model analysis completely redefined the configurations.**

L293, Figure 5: After referring to Fig 3, please try to make it clear here that you are back to referring to Fig 5 here. E.g. "The large variability around those mean relative changes (Fig 5, error bars) ..." Also state in Fig 5 caption what do error bars represent.

**Figure 5 does no longer exist in the revision.**

L297: Yet, I'd somewhat expect larger aboveground woody biomass could also result in bigger trees -> less understory PAR. Does it imply problems in model structure?

**It definitely relates to problems in the model structure (the woody tissues do not account for light interception). It is now discussed in the revision:**

**"...more woody biomass does not translate into exacerbated light interception."**

L329: Unfortunately, there is no Table 5.

**We are sorry that this table somehow disappeared during the submission process. It was included in the new version as Table 6.**

L331-334: IC-TLS uses both the DBH distribution and the allometric coefficients informed by TLS. So I assume it was able to capture Figure3 - leaf biomass relationship very well? Sounds like it also produced LAI values in the right ballpark. The link between leaf biomass and leaf area is through the SLA in the model (L238: SLA is used to convert the leaf biomass into leaf area), and SLA posterior of IC-TLS is agreeing with the CWM. So it almost looks like this configuration gives the right answers for the right reasons for these variables and parameters. And if IC-FC is producing the same leaf area as IC-TLS but have different SLA, it

must be missing leaf biomass? I'm trying to see if authors could discriminate a bit more explicitly about performances of different configurations here, please consider elaborating as such.

**In the new version of the manuscript, we made clear which configurations perform better for every single independent observation (eddy covariance fluxes, leaf area).**

L343: Does this contradict or how is it related with the finding mentioned before on L281-282: when NBG is concerned model structure had a bigger impact than allometries on tree size distribution? Also although I couldn't see Table 5, I have a feeling that it will be hard to digest. I recommend authors consider a figure instead or in addition.

**This sentence (L343) does not contradict previous findings because in that case we were investigating the allometric parameters only and we did not compare the relative process and parametric errors. This is something we now investigate in the revised manuscript. The figures and tables changed significantly but we tried to keep them digestible, as suggested.**

L381-382: Could you be more specific here? The statement is too vague. NBG with TLS informed allometry didn't do any better for capturing the tree size distribution (they also did bad for the ecosystem variables L346). So informing allometry alone was not enough. Is TLS more useful when it is used for prescribing the initial conditions or what? How does this agree with studies in the literature? E.g. does this mean initial condition uncertainty is a bigger problem than allometric uncertainty?

**In the new version, we were able to demonstrate the benefits of TLS to more adequately represent the modelled ecosystem. To do so we compared the performance of the different tested configurations against independent validation datasets (see previous comments). By partitioning the overall uncertainty, we also quantified the relative**

**benefits of prescribing the initial conditions, the model structure and the allometric equations (see Figure 4 in the manuscript).**

L385-386: I don't know if it is striking, but it was expected given the trade-off and uninformative priors.

**This problem was solved by informing the priors with the trait data in the new analysis.**

L388-389: "Very different" but how? Again very vague. Was a particular one any better?

**This now clearly emerges from the new analysis (TLS + TRY constraints), see Table 6 and Figure 7 as well as the corresponding pieces of text.**

L395-396: But did it in this study? Does this mean the authors trust IC-TLS posteriors on Fig 7 more? Also please see my comment above for lines 331-334, and try to be more specific. If you did discriminate between equifinal model versions, say it here which did better.

**We now discriminate between equifinal model versions in the revised manuscript (see previous comments).**

L407: But aren't there more formal ways to deal with this? E.g. one could start the model from the past (when flux data is available) with more uncertain IC even if they don't know about the forest structure and composition a decade ago, and then calibrate the model with past data, continue simulations in time and assimilate more recent inventory data to constrain the states? In fact, the Thomas et al 2011 paper cited by the authors already have some useful values for conditions 10 years ago. Furthermore, the Butt et al. citation implies there was a tree census in 2008? (I merely clicked the link)

**Unfortunately we could not access the raw data of previous tree inventories. As for the time lag between the flux data observations and the inventories/TLS, we are aware of this limitation that we discuss in the paper (first paragraph of the study limitations subsection). In a nutshell, we assume that in 8 years time, the forest composition and tree**

**size distribution should not have changed dramatically enough to alter the main conclusions of this study that address the sources of uncertainty of the ED2.2 model more than the link between functional ecosystem composition and land fluxes.**

L412: While it is true that it would increase the overall complexity of the study, I'm not sure it sufficiently justifies simulating one PFT when at least Acer and Quercus are concerned. Could more informative priors be chosen when more distinct PFTs are used? There were also numerous occasions mentioned above where using multiple PFTs could potentially remedy some of the shortcomings. At least without demonstrating it, I'm afraid this argument remains unconvincing.

**We now solved this issue by (i) testing more than one PFTs and (ii) using more informative priors for some of the traits (see comments above)**

L418: This statement, although true, seems rather irrelevant for the conclusion of the present study as both SLA and Vcmax are measurable. In general, until the last three sentences, the conclusion reads like an introduction and needs to be tailored towards the study more. I'd recommend starting from what you demonstrated, then telling what the implications of your findings are, how well your results aligned with your prior expectations, if your methodology was adequate, if you got new insights / new ideas for future steps and so on.

**We tried to make sure the conclusion was tailored towards our findings, by completely revising it.**

L422-425: Apologies for repeating myself but I overall think the reporting was rather inconclusive as to whether the TLS informed model was indeed more reliable or able to discriminate between equifinal model versions. In other words, yes, TLS-informed results were different but were they more realistic? What was the independent validation? Which

configuration got the right answers for the right reasons? Reader has to work really hard to figure it out. You could further provide your concluding recommendation regarding how TLS is best utilized.

**All those points should be hopefully addressed by the new conclusions emerging from the global sensitivity analysis.**

**Response to the Executive Editor comment**

Dear authors,

We have checked your manuscript, and unfortunately, at the moment, it does not comply with our 'Code and Data Policy'. Currently, you archive the scripts that you use in Github. However, as we state in our policy and Github on its website, it is not a suitable repository for long-term archival.

Therefore, please, move your code to one of the suitable repositories that we list before the end of the Discussions period and make the necessary changes in the manuscript in potential reviewed versions. Be aware that failing to comply with these rules will prevent your manuscript from being considered for publication. https://www.geoscientific-model-development.net/policies/code_and_data_policy.html#item3 Also, you have included the link to Github of the ED-2.2 model, however, you must cite the corresponding Zenodo repository, as again Github is not a secure repository. The Zenodo repository for ED-2.2 is: https://doi.org/10.5281/zenodo.3365659. Please, remember using the corresponding DOI to cite it in the text.

Best regards,

Juan A. Añel, Geosc. Mod. Dev. Executive Editor

Dear Juan A. Añel,

We would like to apologise for not complying with the code and data policy of GMD. The new version of the manuscript includes the links and DOIs of the Zenodo repositories for both the ED2 model and the scripts and data that are necessary to repeat the analyses.

On behalf of all co-authors,

Félicien Meunier

---

## Author Response (AR2)

**Reviewer #1**

Thank you for the revision of this study. This version of the manuscript has seen much thought and improvements. Particularly, the combination of various uncertainties to GPP, LAI, AGB, PAR variances are informative and are presented well in the pie charts. Essentially the study investigates the contribution of process, parameterisation and initialization errors on carbon flux predictions mainly, using a combination of TLS and non-TLS specifications. This is worth pursuing and much of the information is already there, but the study could be better structured to best show the power of TLS in constraining ED2.2 simulations at Wytham Woods.

We thank the reviewer for both the positive evaluation of our in-depth revisions and more generally of the quality of our work. We were glad to read that the reviewer had seen much improvement between the two versions of the manuscript. We also take this opportunity to thank the reviewers from the previous round whose suggestions led to this revised version and greatly improved the quality of this work.

Essentially, the error reductions not directly associated with TLS can be presented separately first and then the different TLS components can be presented more clearly. Currently the information is there but it gets lost in the complexity of results displayed and presented.

There are indeed two ways to present the results of this study. The first separates the error reductions associated or not with TLS, and presents them sequentially (the option suggested by the reviewer). The second mixes them and discusses the sources of error reductions all together (the option we had chosen for the previous version of the manuscript). For our revised version of the manuscript we decided to go for an intermediary option. We would indeed like to keep the illustrations (e.g. Figures 3 and 4) and the results section structure as is but edited the presentation of the results, e.g. to present the reduction of the uncertainty caused by prescribing the initial conditions and constraining parameters with TRY data on the one hand and quantify the added value of using TLS on the other, as suggested by the reviewer.

The reasons why we kept the illustrations and most of the results section structure as in the previous version are multiple. First, we would like to point out that the current way we present the results (a grid of subplots/pie charts with the different initial condition and TRY constraint configurations as in Figures 3 and 4) encompasses the suggestion of the reviewer and even exceeds it. At the end of the day, the error reduction not directly associated with TLS is the change of the variance (and of its components) going from the "NBG-without TRY constraints" to the "Census-with TRY constraints" configuration. While the added value of TLS to the model configuration with the lowest uncertainty can be grasped from the change of uncertainty (and of its components) from the "Census-with TRY constraints" to the "TLS-with TRY constraints" configuration. So comparing specific pairs of pie charts or subplots in Figures 3-4.

Secondly, presenting the outputs from all model simulations together allows the reader to compare (i) the model performance and uncertainty (e.g. Figure 3) and (ii) the model variance and its components (e.g. Figure 4) directly between these specific pairs of configurations as well as others, which might be relevant in some cases. In this study, we had indeed the chance to have close-in-time inventory data, TLS-derived allometries and size distribution, species-level trait data from TRY and eddy-covariance data (or rather we selected the site for that reason). Yet, field inventories and/or trait data are not always available to reduce the initialization and the parameter errors. Therefore, presenting together the benefits of adding different data sources enables the comparison between the specific configuration that an interested reader could experience for his/her site as well as the potential added value of TLS for that site.

Finally, in our opinion it is important both from a theoretical and a practical point of view to weigh the relative and absolute benefits of different data sources to reduce the model uncertainty and increase model performance. Theoretically first, identifying the most important sources of model uncertainty provides insights and a better understanding of the functioning of such a nonlinear model. Our presentation also allows one to directly compare the benefits of TLS with respect to other data sources and hence its comparative value compared to other observation types, which might be important for field campaigns that are time- and resource-limited. Presenting the error reduction first with inventory and trait data and then from that configuration to trait + inventory + TLS would make our study less systematic and comprehensive.

Yet, we agree with the reviewer that the suggested way of presenting the results also has its advantages. Therefore, in the new version of the manuscript, we added more direct description of the error reduction from the "NBG-without TRY constraints" to the "Census-with TRY constraints" configuration (i.e. the error reduction not associated with TLS), from the "Census-with TRY constraints" to the "TLS-with TRY constraints" configuration (i.e. the added value of TLS with respect to the configuration with the lowest uncertainty), as well as from the "NBG-without TRY constraints" to the "TLS-with TRY constraints" constraints" configuration (i.e. the added value of TLS with respect to the configuration with the lowest uncertainty), as well as from the "NBG-without TRY constraints" to the "TLS-with TRY constraints" configuration (i.e. the added value of TLS with respect to the configuration with the lowest uncertainty). We added/adapted the following paragraphs in the results section:

"When both parameters were constrained and realistic initial conditions were prescribed to the model (i.e. going from the NBG-without TRY constraints to the Census-with TRY constraints configuration), the variability of the simulated GPP experienced a three-fold decrease. Similarly, the variability of LAI (supplementary Figure S6-7) and AGB (supplementary Figure S8) was drastically reduced, with a four-fold and and a two-fold decrease respectively."

Given the similarities of the tree size distributions derived from the inventory and TLS (see results section 3.1), prescribing initial conditions had a similar impact on the variability of the outputs for the TLS and for the Census configurations. Combined with the constraints on allometries, it led to a reduction of the ensemble standard deviation for GPP in June to 3.78  $\mu$ mol m-2 s-1 for the TLS configuration without TRY constraints. As for the Census configuration, constraining SLA and Vc,max with TRY data had a larger impact on the model uncertainty: ensemble standard deviation of GPP in June for the TLS configuration with TRY constraints decreased to 1.54  $\mu$ mol m-2 s-1."

"All in all, the predicted variability of the ecosystem LAI and GPP was the lowest for the TLS configuration with TRY constraints:  $3.79 \pm 0.50 \text{ m}^2 \text{ m}^{-2}$  for the ensemble mean (± one standard deviation) of the ecosystem LAI (Supplementary Figure S6), 9.86 ± 2.89 µmol m-2 s-1 for the ensemble mean (± one standard deviation) of the ecosystem GPP (Figure 3), both

during leaf-on conditions, which compared well with independent observations (Table 6). The confidence interval of the simulated ecosystem GPP in June for the TLS configuration with TRY constraints was significantly reduced (11.8 - 17.6  $\mu$ mol m-2 s-1) and much closer to the confidence interval of the observations (11.5 - 14.6  $\mu$ mol m-2 s-1).

Reducing parameter and process errors is crucial for TBM studies. In this study the purpose of doing this is to investigate component errors, yes, but also to present a model configuration that has the lowest uncertainty before implementing TLS related constraints.

This is probably where we disagree with the reviewer. Why would we necessarily need to examine the added value of TLS as compared to the configuration with the lowest uncertainty? In other words, what should we only consider the benefits of TLS-related constraints on top of all other data sources, and not rather also in replacement of a part or of the totality of these other data sources? Modellers can sometimes find themselves with no data sources (no inventories, no field data) to reduce the model errors. Our presentation provides an estimate of the benefits of TLS scanning in such situations, together with the benefits of incrementally adding TLS data to trait data, field inventories and both. We also added in the results section:

"Incrementally adding the TLS-related information to the Census-with TRY constraints configuration had a positive, yet more limited effect on the reduction of the model variability of GPP: ensemble standard deviation of GPP in June was reduced by 30% between the Census and TLS configurations with TRY constraints. Constraining allometries with TLS had a more significant impact on LAI (supplementary Figures S6-S7) and AGB (supplementary Figure S8), with a three-fold decrease of the ensemble standard deviation from the Census-with TRY constraints to the TLS-with TRY constraints configurations."

and

"In total, the variability of the simulated GPP experienced a four-fold decrease when parameters were constrained, realistic initial conditions were prescribed, and TLS data were used to constrain the allometries (i.e. going from the NBG-without TRY constraints to the TLS-with TRY constraints configuration)."" Therefore, perhaps it is better to present the process errors first (crown size / RTM / PFT) for the NBG and census simulations. Second you would present the parameter errors (from Table 4). Here you can present the TRY parameters including all other parameters from Table 4. What will come out of this is a) the contribution of these parameter and process errors to ED2.2 carbon flux variance, and b) The model configuration that more closely matches GPP observations (and LAI/AGB/PAR).

Once you have achieved this, you will delve into the TLS benefits to constrain structure and function in ED2.2, as stated in the title and in your study aims. The TLS benefits are a) Initial ecosystem structure – INITIALIZATION ERROR; b) Allometry (bleaf, tree height, AGB, crown area) – PARAMETERISATION ERROR components; and c) SLA, reflectance, clumping – PARAMETERISATION ERROR components. You can present each of these components combined, or separately – or as part of the pie chart format you created. Currently allometry (dark green) and initialization error (inferred in circle size) are in Figure 4 (S7-S9). It would be good to separate these from the process errors and other errors not directly linked to TLS improvements. This way, the improvements from TLS can be clearly visible.

For the first suggestion, we refer to our previous two responses. Yet, we would like to point out that we modified the order of the sentences to follow reviewer' suggestion to first go from error reduction using IC and trait data, to error reduction from IC + trait data to IC + trait + TLS.

Regarding the TLS benefits, as we describe in more detail below, there are only two (initial ecosystem structure and allometry) - not three - that can currently be considered in this study since theoretical, technological, and technical challenges still exist to derive parameters like clumping and reflectance from TLS data.

The pie charts we present allow one to grasp the benefits of using TLS to reduce both initialization and the parameterisation errors from different situations (when field inventories are available or not, when trait data exist or not). We disagree with the reviewer that we should separate the benefits of TLS from other errors not directly linked to TLS improvements for the reasons mentioned above (e.g. to easily compare the relative

**benefits of TLS vs other data sources starting from multiple situations for multiple outputs and multiple error types).**

In terms of the third component above (SLA, clumping and reflectance (?) from Table 4), I do not think this has been considered yet. It may be worth thinking about quantifying these parameters using TLS at Wytham Woods and them determining their potential to constrain ED2.2. If not possible for this study, you could cover the error reduction you might expect from TLS. All in all, this separation of error components will very clearly show the improvements in model performance TLS data can have.

In recent years, TLS has been indeed used to extract more and more information about canopy and plant structural traits, next to the allometries. Yet, those studies remain as of today exploratory and the methods supporting them are either under development or still suffer from important drawbacks. Therefore, we cannot apply them directly to our TLS dataset to derive additional traits like SLA, clumping, leaf angle distribution or reflectance for our study site. It might be possible in the future though. Therefore we added (L461-466, see new manuscript version with no track change):

"In the future, TLS could inform vegetation models even more. The TLS community is indeed actively working on the derivation of additional tree- or stand-scale parameters from lidar raw data and 3D point clouds. Those parameters include leaf angle distributions (Boni Vicari et al. 2019), clumping (Zhao et al. 2012), and reflectance (Calders et al. 2017), which have been shown to significantly contribute to the overall model uncertainty (Meunier et al. 2021; Shiklomanov et al. 2020; Viskari et al. 2019). Yet, theoretical, technological, and technical challenges specific to each parameter still need to be raised before one can constrain these sensitive traits with TLS in a study similar to this one."

Also: Why not present GPP evaluations for the whole 2 year period?

There are two main reasons why we do not present the evaluation against GPP data for the full 2 years period. First, the year-to-year variability in the dataset is extremely low (see the original publication) and therefore so is the added value of using the years separately rather than aggregated. Second, we could not access local met drivers to force our model simulation and used gridded meteorological forcings instead (CRU-NCEP). Doing so, the model cannot capture small timescale variations or day-to-day variability. Yet, it can (and does) reproduce the observed seasonal cycle and that is what we decided present here. We added in the discussion:

"In addition, in the absence of locally observed meteorological drivers, we had to force the model simulations with regional datasets that cannot serve the purpose of capturing the day-to-day variability or the diel cycle, which forced us to only compare the modelled and observed seasonal GPP cycle."

Are you sure the calculations of NEP are correct in table 6? The seem much lower than the difference between GPP and Ecosystem respiration.

The confusion probably comes from the fact that the GPP presented in Table 6 is for the leaf-only period why ecosystem respiration and NEP are for all year round, as indicated by the superscript (1). We re-designed Table 6 and added all flux variables for both the leaf-on period and all year round to make it more clear.

**Reviewer #2**

Revised submission by Meunier et al. addresses the previous reviewer concerns to a great extent. Re-organizing and re-focusing the paper made it clearer, results are now more intelligible and conclusive. I believe it will be a valuable reference for both the TLS and the modeling (especially ED2) community. I recommend its publication with the following minor comments (page and line numbers refer to the revised un-tracked manuscript):

We thank the reviewer for the very positive feedback and once more for the suggestions from the previous round that significantly improved the quality of the manuscript.

p4.l81: "would be" could be removed?

**We removed "would be"**

p5.I5 / p16.I14: This is probably a matter of taste but to me, fusing data and models have a more formal statistical meaning where multiple data sources are formally combined in a data assimilation algorithm or a framework that encapsulates DA and analyses around it towards generating a synthesized data product. Whereas here, the authors are merely informing individual parts of the model independently. Therefore, I would prefer terms like "to inform" or "to constrain" (like authors also say in many parts of the text, e.g. p4.I86) instead of "to fuse". In the end, I leave it to the authors but I wanted to point it out.

We replaced the three instances of "fuse/fusion" in the previous version of the manuscript with other terms like "to inform" or "to constrain".

p7.I52: Could you state what this default value for the temperature coefficient Q10 is already here?

**Added. The new text reads (L155):**

**"(...) using the model default value for the temperature coefficient Q10 of 2.4"**

p8.L83: Could you state what this dominant soil type is already here?

**Added. The sentence now reads (L185-186):**

**"Soil texture was set according to the dominant soil type (clay), (...)"**

p9.199: Could you be a little more specific here, was the trait meta-analysis run only with TRY data or including TRY data? In other words, was there any other additional data informing this analysis besides TRY?

The analysis was performed with data from TRY only. We now emphasise this in the new version of the manuscript L203:

**"The meta-analysis was informed by TRY data only"**

p10. I39 Could you elaborate on how the uncertainty of data was accounted for in this Bayesian analysis?

We fitted the allometric models using all the available data and the 'brms' package, which generates the posterior distribution of the allometric parameters. To account for the uncertainty of the data, we used a bootstrapping method. We added a description of such a method (L241-244):

"More specifically, we fitted the parameters of the four allometries of ED2.2 using a Bayesian approach and the 'brms' package of R (Bürkner 2017). To account for the uncertainty of the data, we repeated the same analysis multiple times (N = 100) using random sampling with replacement and aggregating the resulting allometric parameter posterior distributions."

Table 6 caption: Would it be a bit more accurate to say "states and fluxes" as one wouldn't really call GPP/NEP/resp states.

We agree with the reviewer. We replaced "state variables" with "states and fluxes" (see new caption of Table 6).

p14.138: Was the meta-analysis run with random effects turned off? See Raczka et al., meta-analysis posteriors can be too narrow when that is the case, please acknowledge this here or in the discussion.

Indeed the meta-analysis was run with random effects turned off. We added it in the analysis description (L202) and now discuss this additional limitation (L501-503):

"Third, the trait meta-analysis was run with random effects turned off, which can generate too narrow parameter posterior distributions (Raczka et al. 2018), and hence underestimate the contribution of TRY-constrained parameters (see e.g. Figure 4). A similar analysis including random effects should be repeated to evaluate such an underestimation."

p16.l01 Please correct the typo: use "satisfactorily" or "to satisfactory levels"

**Corrected.**

p.16.103-06: Looking at Table 2, in general, I see results and discussion on many of the configurations listed here, except the trait plasticity. This could be a topic of interest for the readers, could the authors report more on it in the manuscript?

We agree with the reviewer and now added the following description of the results (L371-L373):

"Process uncertainty was dominated by the type of crown model (5%) and the radiative transfer model (4%). Trait plasticity only contributed marginally to the overall variance (< 1% on average)."

p17.l17 Please correct the typo: delete "is"

Corrected.

---

## Author Response (AR3)

Dear authors,

The new version of the manuscript addresses well the comments from the reviewers. However, this version is missing a code and data availability statement. Please follow GMD's policy on code and data availability (https://www.geoscientific-model-development.net/policies/code_and_data_policy.html), and submit a revised version with the corresponding links.

Regards,

Carlos A. Sierra

**We moved the sentences related to the code and data availability to a specific section, located just before the acknowledgements, which reads:**

**"Code and supporting data (including initialization and setting files) for reproducing the results presented below are publicly available in Zenodo and have the permanent DOI 10.5281/zenodo.6363617. The ED2.2 model is available at https://doi.org/10.5281/zenodo.3365659."**

**Best regards,**

**On behalf of all coauthors,**

**Félicien**

---

## Author Response (AR4)

1. Your tables contain coloured cells or/and coloured values. Please note that this will not be possible in the final revised version of the paper due to HTML conversion of the paper. When revising the final version, you can use footnotes or italic/bold font. But if the colour spectrum is necessary and cannot be exchanged for footnotes, bold, or italic, then please inform us via email.

2. Please ensure that the colour schemes used in your maps and charts allow readers with colour vision deficiencies to correctly interpret your findings. Please check your figures using the Coblis – Color Blindness Simulator (https://www.color-blindness.com/coblis-color-blindness-simulator/) and revise the colour schemes accordingly.

**1. In the new version of the manuscript, I removed the coloured cells**

**2. I checked the figures using the Coblis-color blindness simulator and it looks like they allow readers with colour vision deficiencies to correctly interpret our findings.**

**3. I added the author contribution and competing interest sections that were missing**